# Thinking Sparks!: Emergent Attention Heads in Reasoning Models During Post Training

## Abstract

The remarkable capabilities of modern large reasoning models are largely unlocked through post-training techniques such as supervised fine-tuning (SFT) and reinforcement learning (RL). However, the architectural mechanisms behind such improvements remain largely opaque. In this work, we use circuit analysis to demonstrate that post-training for complex reasoning sparks the emergence of novel, functionally specialized attention heads. These heads collectively support structured reasoning and computation. Our comparative analysis across Qwen families and Qwen-based DeepSeek-distilled model reveals that these emergent heads evolve differently under different training regimes. Distillation and SFT foster a cumulative addition of stable reasoning heads. In contrast, group relative policy optimization (GRPO) operates in a dynamic search mode: relatively few attention heads are iteratively activated, evaluated, and pruned, with their survival closely tracking fluctuations in the task reward signal. Furthermore, we find that controllable "think on/off" models do not possess dedicated "thinking" heads. Instead, turning off explicit reasoning triggers a broader—but less efficient—set of compensatory heads. Through ablation and qualitative analyses, we connect these circuit-level dynamics to a crucial performance trade-off: strengthened heads enable sophisticated problem-solving strategies for difficult problems but can also introduce "over-thinking" failure modes, such as calculation errors or logical loops on simpler tasks. These findings connect circuit-level dynamics to macro-level performance, identifying an inherent tension where complex reasoning comes at the cost of elementary computations. More broadly, our work points to future directions for training policy design, emphasizing the need to balance the development of effective reasoning strategies with the assurance of reliable, flawless execution.

## 1 Introduction

The advent of large reasoning models (LRMs), such as OpenAI o-series (Jaech et al., 2024; OpenAI, 2025b) and DeepSeek-R1 (Guo et al., 2025), has marked a significant milestone in artificial intelligence, demonstrating unprecedented ability in solving complex, multi-step problems. These models typically employ Chain-of-Thought (CoT) process (Wei et al., 2022b), generating an explicit sequence of reasoning steps before arriving at a final answer. This capability is substantially enhanced by extensive post-training methods, primarily supervised fine-tuning (SFT) and reinforcement learning (RL) (Trung et al., 2024; Xi et al., 2024; Mukherjee et al., 2025), and by allocating more test-time compute during inference (Zhang et al., 2025b; Wu et al., 2025b; Snell et al., 2025).

Despite their empirical success, the mechanisms by which these methods enhance reasoning remain largely unclear. This opacity presents a significant

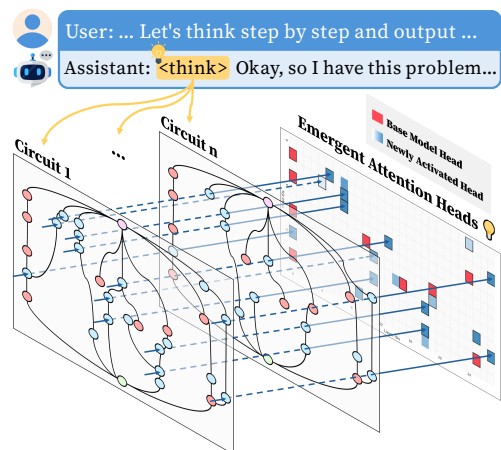

Figure 1: Reasoning circuits trace the internal computations of LRMs at each checkpoint. After post-training, newly activated attention heads influence the performance at those checkpoints.

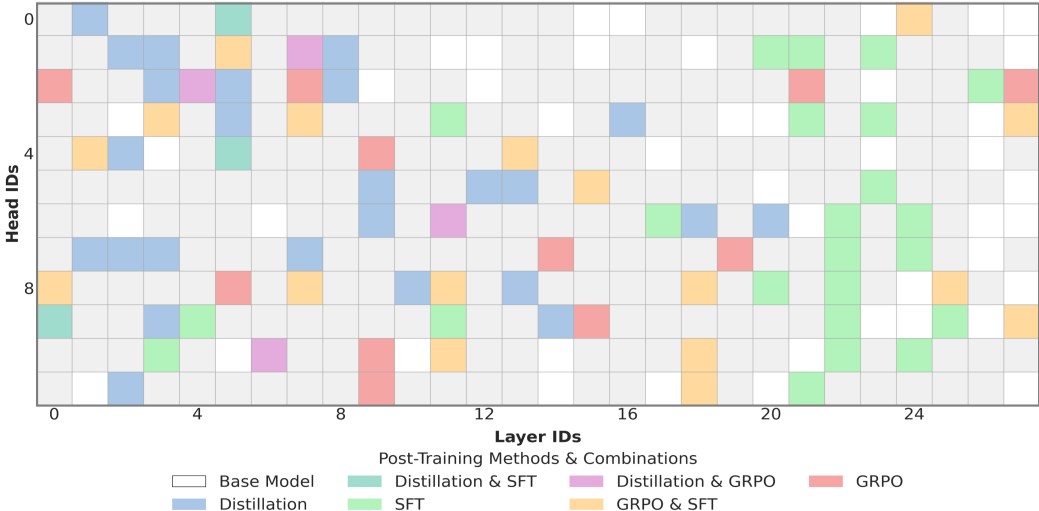

Figure 2: Visualization of emergent attention heads in circuits based on Qwen2.5-Math-1.5B with various post-training, and DeepSeek-R1-Distill-Qwen-1.5B. Each GRPO and SFT category encompass both AIME and AMC benchmark based circuits, with checkpoints of both training using OpenR1-Math-220k and GSM8k dataset. DeepSeek Distillation activates enormous heads (blue), as SFT activates similarly large amount of heads, though SFT heads are mostly concentrated in mid-to-late layer (green). Some of heads from GRPO training are also common in the SFT and Distillation reasoning heads (yellow and purple), however, the number of GRPO heads are much smaller and distributed across layers (red). Comparison with layer wise cumulative map is in Figure 15.

challenge. For instance, post-trained models often suffer from the "overthinking problem" (Chen et al., 2024; Sui et al., 2025), generating excessively long and computationally expensive reasoning chains even for simple tasks, which highlights a critical need for more efficient and adaptive strategies (Tu et al., 2025; Zhang et al., 2025c). Furthermore, the community lacks a clear understanding of the fundamental differences between post-training paradigms. Recent studies have debated whether these methods instill genuinely new problem-solving skills or merely amplify latent capabilities already present in the base model (Rajani et al., 2025; Yue et al., 2025; Ma et al., 2025). Motivated by these trade-offs, several works have proposed "Think On/Off" controls to manually modulate reasoning depth (Wu et al., 2025a; Yang et al., 2025; OpenAI, 2025a). However, without a granular understanding of how post-training alters a model's internal mechanism, efforts to improve reasoning are confined to trial-and-error adjustments of training data and resources (Mukherjee et al., 2025).

In this work, we bridge this gap by shifting the analysis from high-level performance metrics to a low-level mechanistic investigation of the model's internal workings. We employ circuit analysis, a powerful tool of mechanistic interpretability, to identify and characterize functional subgraphs within the transformer architecture (Vaswani et al., 2017) that are responsible for specific behaviors (Olah et al., 2020; Elhage et al., 2021; Wang et al., 2023; Bereska & Gavves, 2024; Lindsey et al., 2025). By applying these lens, we trace the formation of specialized groups of attention heads through reasoning circuits that emerge as a direct consequence of post-training procedures. This direction is motivated by preliminary findings that particular attention heads correlate with the quality and length of a model's reasoning (Voita et al., 2019; Cabannes et al., 2024; Reddy, 2024).

Our investigation and ablation study yield a series of clear, mechanistically insightful findings:

1. Distillation and SFT: We find that distillation and SFT induce a large amount of newly emergent heads in circuits. Distillation heads are mostly found in early-mid layers, whereas SFT heads are focused on mid-to-late layers. They effectively instill complex reasoning with a considerable proportion of attention heads, which also have a potential of confusion.

2. Group Relative Policy Optimization (GRPO): A prominent RL algorithm, GRPO, engages in dynamic search for reasoning attention heads during the training process, mirroring fluctuations of the task reward signal. Its targeted, minimal, but high-impact edits optimize the use of existing knowledge and computational pathways, not building entirely new ones.

3. Thinking On/Off: While think on mode does not have its own exclusive reasoning heads, think off mode activates enormous attention heads to compensate performance gaps. Disabling or scaling down those thinking off heads temporally boosts its performance, but those heads are crucial asset for robust problem solving when the sampling coverage increases.

## 2 PRELIMINARY

Transformer circuit models the internal computation of its architecture as a directed acyclic graph (DAG) $G = (\mathcal{N}, E)$, where $\mathcal{N}$ is the set of circuit nodes and a generic node is denoted by $n \in \mathcal{N}$. Each node corresponds to a distinct component in the model: attention heads $A_{l,j}$ (at layer $l$ and head $j$), MLP modules $M_l$ for each layer, the input node $I$ (embeddings), and the output node $O$ (logits), following (Nanda et al., 2023; Conmy et al., 2023; Ameisen et al., 2025):

$$\mathcal{N} = \{I, A_{l,j}, M_l, O\}. \tag{1}$$

Edges $E \subseteq \mathcal{N} \times \mathcal{N}$ encode how each node's output contributes to later layers' residual stream inputs:

$$E = \{(n_x, n_y) \mid n_x, n_y \in \mathcal{N}\}. \tag{2}$$

A circuit is defined as a subgraph $C \subseteq (N, E)$ selected to explain a specific behavior, e.g, how certain tokens influence the model's output or how factual knowledge is stored and elicited (Yao et al., 2024a; Ou et al., 2025; Park et al., 2025). We specifically implement edge attribution patching with integrated gradients (EAP-IG) which improves faithfulness, wherein ablating all non-circuit edges preserve task performance (Nanda, 2023; Hanna et al., 2024).

Let $(n_u \to n_v) \in E$ and let $z_u$ and $z'_u$ denote the clean and corrupted activations of node $n_u$'s output into the residual stream, respectively. We define the input difference along this edge as $\Delta z_u = z_u - z'_u$. Following the integrated gradients rule, we average gradients along the straight-line path from $z'_u$ to $z_u$. As the scalar output signal, we apply a task-agnostic divergence $\mathcal{L}(y_{\text{clean}}, y)$ between the model's output logits at the target position under the clean and interpolated activations, typically a KL divergence. We then take gradients of this scalar signal with respect to the *input of node $n_v$* (i.e., $n_v$'s pre-activation into the residual stream). The EAP-IG edge score is

$$\text{score}(u \to v) \;=\; \Delta z_u \cdot \frac{1}{m} \sum_{k=1}^{m} \left. \frac{\partial \mathcal{L}\big(z' + \frac{k}{m}(z - z')\big)}{\partial (\text{input of } n_v)} \right|_{z' + \frac{k}{m}(z - z')}, \tag{3}$$

where $m$ is the number of Riemann-sum steps approximating the IG path integral. We rank edges by equation 3 and select a sparse set by *top-n* selection. Lastly, we prune isolated nodes and validate faithfulness via post-hoc interventions: ablate all non-circuit edges (e.g., patching to baseline) and check that task performance is preserved. Detail of scoring is in § A.3. Beyond the scope of our main analysis, Sparse Feature Circuits (Marks et al.) are further examined in §A.5 and Figure 14.

## 3 IDENTIFYING EMERGENT ATTENTION HEADS WITH CIRCUITS

To systematically compare how different post-training paradigms change a model's internal mechanisms, we design a rigorous experiment based on circuit analysis. Our methodology focuses on identifying and validating the causal roles of emergent attention head circuits. Details of the experimental setup are provided in Appendix A.1.

### 3.1 CIRCUIT DISCOVERY & ABLATION INFERENCE

Our core methodology for identifying emergent reasoning circuits is a practical application of causal analysis, using ablation as a proxy for more complex patching experiments. The process is as follows:

**Circuit Mapping.** For a given task (e.g., solving an AIME problem), we first map the active computational graph for both the baseline model and a post-trained model. As the circuit is structured with pairs of prompts, clean and corrupted, we set clean prompts designed to elicit the reasoning behavior by sampling the answer of each model category.

- Baseline model: Answers such as "To determine the molecular ..." or "We'll use Python to help us solve ..." for clean, while reasoning model's answer become corrupted.
- Reasoning model: Answers right after <think> such as "Okay, so I have this problem ..." and "Alright, so I need to find ..." for clean, while baseline model's answer become corrupted. Samples can be found in §A.2.

Table 1: Reasoning Head Ablation Inference for DeepSeek-R1-Distill-Qwen-1.5B and 7B. Every performance is measured with pass@1 score with temperature 0.6 and 32k context length. Each ablation cases make the value of specific attention heads, around 5 to 10 number of heads from its circuit results, into zero for checking its importance for reasoning tasks. We color some scores into red which is the most degraded results except no ablation baseline, while the bold is the completely ruined performance. We also color performance increase with green when its heads are ablated.

| Model | Method | AIME'24 | AIME'25 | GPQA | AMC |
|---|---|---|---|---|---|
| DeepSeekR1-Distill Qwen-1.5B | No Ablation | 30.0 | 26.6 | 18.6 | 66.2 |
| | Ablation with Reasoning Heads | 26.6 | 16.6 | 17.1 | 59.0 |
| | Ablation with Base Model Heads | 30.0 | 23.3 | 12.1 | 53.0 |
| | Ablation with TriviaQA Heads | **0.00** | **0.00** | **0.00** | **0.00** |
| DeepSeekR1-Distill Qwen-7B | No Ablation | 40.0 | 43.3 | 35.3 | 81.9 |
| | Ablation with Reasoning Heads | **53.3** | **46.6** | 35.8 | 78.3 |
| | Ablation with Base Model Heads | **53.3** | 43.3 | **37.3** | **83.1** |
| | Ablation with TriviaQA Heads | 50.0 | 50.0 | 34.3 | 79.5 |

**Identifying Emergent Components.** By comparing circuits of the post-trained model to that of the baseline model, we identify the set of "emergent heads"—those that are active in the post-trained model but not in the baseline. These heads represent the structural changes induced by the training process. Basically, we specifically pick Qwen families for pair comparison. We also re-implement our approach on the Llama-3.2-1B-Instruct (Meta, 2024b), applying two distinct post-training methods: SFT and GRPO, for more generalizability. In addition, further importance based analysis in §A.4 and Figure 13 is qualitatively support our basic emergence based differentiation.

**Causal Validation via Ablation.** To confirm that these emergent heads are causally responsible for the new reasoning capabilities, we perform ablation inference. We run the post-trained model on the evaluation benchmarks but surgically disable the emergent heads by zeroing out their outputs. A difference in performance on the target task, compared to the intact post-trained model, serves as strong causal evidence that these heads form a critical part of the newly acquired reasoning circuits.

**Head Activation Scaling.** Furthermore, we scale up/down activations of each reasoning head in baseline model with their attention head index (layer num and head num). We then find out the difference in performance both quantitatively and qualitatively.

Figure 1 shows the overall process of our circuit findings. And Figure 16 to 18 visualize circuits.

## 4 In-Depth Analysis on SFT & Distillation

Our investigation reveals that different post-trainings do more than simply fine-tuning a model's parameters—they fundamentally reshape its internal architecture by activating new attention heads.

### 4.1 Distillation heads strongly affect to performance

The primary finding is that distillation induces a set of new, consistently activated attention heads that are not present in the baseline circuits for the same tasks like AIME'24 and AMC, as in Table 3 and Figure 2. Although two-thirds of the attention head nodes and all MLP nodes active in the baseline model remain active in the distilled one as well, the number of these new heads is significant. They represent an addition to the model's existing machinery rather than a complete replacement, indicating that distillation builds upon the pretrained foundation by writing in new, specialized components.

To validate the functional role of these newly identified heads, we perform attention head ablation experiments. We systematically deactivate a set of emergent reasoning heads in the distilled model and measure its performance. The results as in Table 1 demonstrate a consistent degradations in performance across all benchmarks, e.g., AIME'24 pass@1 drop from 30 to 26.6. Although the drop rate is smaller in GPQA and AMC as emergent reasoning heads are usually from the circuits of AIMEs, the degradation remains significant. We also compare their effectiveness against other heads, base model-exclusive heads with same benchmarks and Heads from TriviaQA circuits. Here, as 1.5B model is too sensitive for head ablation like the case of TriviaQA heads, leading to the score of zero, ablating base model heads in 7B model is quite interesting as its overall performance goes up across various benchmarks. This provides a hint that not all attention heads emerging from post-training are important for reasoning, or they can confuse the model when finding the suitable solution.

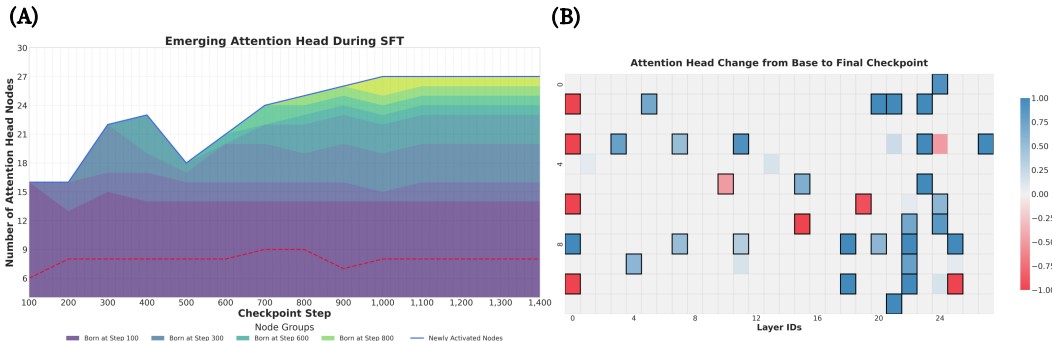

Figure 3: Analysis of Emergent Attention Head in Qwen2.5-Math-1.5B during SFT, trained with OpenR1-Math-220k (Hugging Face, 2025) and circuit construction with AIME AIME (2025). (A) denotes a cohort analysis of attention head activation over training checkpoints. The blue line tracks the absolute number of newly activated heads compared to the base model, while the red dashed line indicates the number of original heads that are maintained. The stacked areas represent cohorts of heads, color-coded by the checkpoint at which they first emerged, showing their persistence and evolution over time. (B) shows a heatmap detailing the changes in activation frequency. Red cells denote heads from the original base model, with fading intensity indicating their gradual deactivation. Blue cells represent newly emerged heads, with darker shades signifying higher activation frequency across checkpoints. Heads active in the final checkpoint are outlined black.

The results of Table 4 further strengthen our insights, as Qwen2.5-Math is more sensitive to ablating its base-model heads than the reasoning heads, reversing the trend seen in DeepSeekR1-Distill of Table 1. This cross-model asymmetry confirms that the heads uncovered by our circuits are specific functional units, rather than a single universal pool of attention heads shared across models.

## 4.2 SFT INTRODUCES LARGE AMOUNT OF ATTENTION HEADS IN MIDDLE-TO-LATE LAYERS

We reproduce a method where SFT is applied to mimic reasoning traces, approximating the effect of distillation. Following §A.1, we train baseline model with OpenR1-Math-220k dataset and construct circuits for each 100 step checkpoints. The results are in Figure 3 and 10. Similar with DeepSeek distillation, SFT-trained models consistently activate a large amount of additional attention heads, and almost every head continuously survives until the training is finished. Half of them emerge at the step 100 checkpoint, and most of them are in middle-to-late layers. This pattern of newly activated heads tending to persist throughout training, indicates the steady construction of new pathways for reasoning in the internals of model.

**Quantitative Analysis.** We conduct ablation same as §4.1 with those many mid-to-late layer's SFT reasoning heads. When we ablate around 10 heads from mid-to-late layer, the performance of every benchmark drops significantly, close to zero. This phenomenon is consistently observed at multiple checkpoints, regardless of their performance. Going further, we also scale up those heads in baseline to check its effectiveness by enhancing their activation 1.3 higher, and it reveals out those heads introduce a trade-off of performance. Although the MATH score increases, the AMC decreases slightly, and the AIME'24 still drops significantly. Detail of performance change is in the Table 5.

**Qualitative Analysis.** When we do a comparative analysis on the newly solved and newly missed problems at each checkpoint, we find meaningful insights into the performance trade-off. After SFT, models try to solve questions in an over complicated way, such as replacing a one- or two-step algebraic manipulation with long substitutions or theory first detours. This leads a net degradation, as the number of newly introduced errors surpassed the number of resolved ones. This shows that, although SFT installs a new, fixed piece of machinery with nudging models toward careful, procedure-following math, it costs strategy selection and path efficiency, causing them to miss previously solved items. Examples of qualitative analysis are in Appendix A.8.1 and A.9.1.

**Re-Implementation for Generalizability.** We do the exactly same approach with Llama3.2 model, and the result is in the Figure 11. Here, overall trend is similar with Qwen's result, as it introduces enormous emergent attention heads, and is cumulative since heads were born at specific checkpoints then stacked across training process. However, the index of emergent heads is quite different as

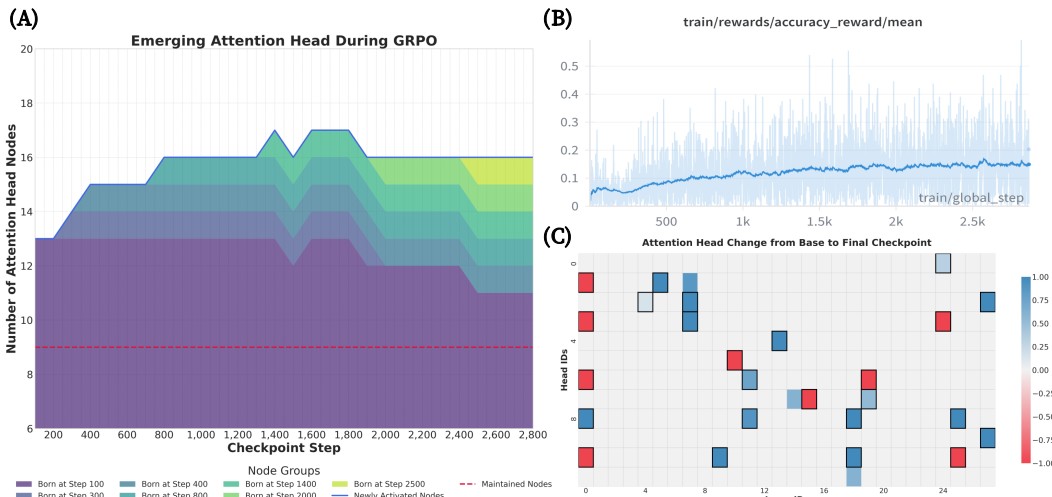

Figure 4: Analysis of Emergent Attention Head in Qwen2.5-Math-1.5B during GRPO, trained with OpenR1-Math-220k (Hugging Face, 2025) and circuit construction with AIME AIME (2025). (A) denotes a cohort analysis of attention head activation across trained checkpoints. The blue line tracks the absolute number of newly activated heads compared to the base model, while the red dashed line indicates the number of original heads that are maintained. The stacked areas represent cohorts of heads, color-coded by the checkpoint at which they first emerged, showing their persistence and evolution over time. The fluctuation in newly activated heads shows a similar trend to the (B), accuracy reward curve. (C) shows a heatmap detailing the changes in activation frequency. Red cells denote heads from the original base model, with fading intensity indicating their gradual deactivation. Blue cells represent newly emerged heads, with darker shades signifying higher activation frequency across checkpoints. Heads active in the final checkpoint are outlined black.

it is not focused on mid-to-late layer. They are widely spread across early-to-mid layers, and we hypothesize that this comes from the different baseline ability between Qwen2.5 Math (already possess certain level of math ability) and Llama3.2 (just instruct model with low calculation ability).

## 5    IN-DEPTH ANALYSIS ON GRPO

**GRPO helps to find the optimal reasoning path.**    In stark contrast to the static installation of emergent heads by SFT, GRPO reveals a dynamic and adaptive process of architectural refinement. Here, emergent heads are not fixed but evolve in response to the reward signal. Like SFT, we train baseline model with OpenR1-Math-220k and this time, also train with another dataset GSM8K shown in §A.1. We construct circuits for each 100 step checkpoints for each dataset version, and the results of AIME'24 circuits are in Figure 4 and Figure 7. Results of different learning rate are in the Figure 8. And circuits with AMC is in the Figure 9. The temporal analysis of GRPO training checkpoints shows that the set of active attention heads is in constant flux. As in Figure 4 (A), the number of newly activated heads rises and falls throughout training, and these fluctuations are strongly correlated with the model's accuracy reward curve while training, as shown in Figure 4 (B). Heads that emerge early in training may be pruned later if they do not consistently contribute to positive rewards, while new heads continue to be trialed throughout the process, even though its overall number is not that many. This suggests an iterative search towards finding an optimal circuit configuration. Notably, the final set of emergent heads after GRPO is small and targeted, and crucially, does not much overlap with the mid-to-late heads by SFT, indicating that the two methods discover different functional specialization.

This dynamic behavior is a direct mechanistic manifestation of the explore-exploit trade-off inherent to reinforcement learning. The activation of a new head represents an exploratory step, which is a test of a new computational strategy. The retention or pruning of that head based on its impact on the reward signal is exploitation, where the model refines its architecture to favor strategies that work.

This circuit-level perspective provides a compelling explanation for why RL acts as a scalpel (Rajani et al., 2025) and results in sparse heads updates. GRPO is not overwriting the model wholesale; it is performing a targeted search for minimal, high-impact edits to the model's functional architecture (Mukherjee et al., 2025) . This also clarifies why RL-trained models' capabilities often remain

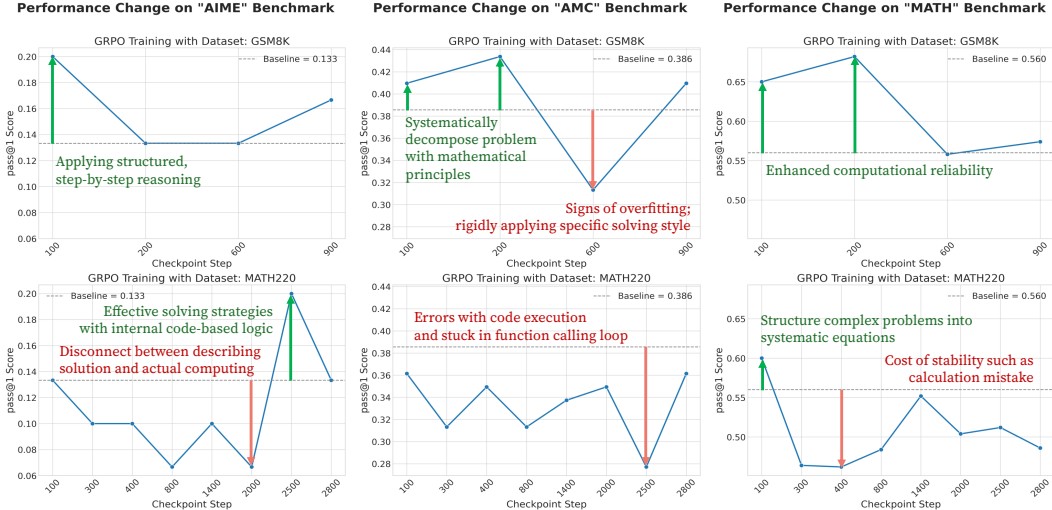

Figure 5: Performance change among various benchmarks for each checkpoints of GRPO training with two different training dataset: GSM8K (Cobbe et al., 2021) and OpenR1-Math-220k (Hugging Face, 2025). The green and red arrow indicate impressive performance gain and lose among various checkpoints, and the captions are the summaries of qualitative analysis. The performance trade-off of each checkpoints is similarly reproduced when we apply attention head scaling with emergent reasoning heads for the baseline model. Actual examples are presented in the Appendix A.8 to A.9.

bounded by the base model's potential (Yue et al., 2025). GRPO is primarily optimizing the use of existing knowledge and computational pathways, rather than building entirely new ones from scratch.

**Quantitative Analysis.** With the similar approach of Section 4.2, we make a difference among the scales of each attention heads. When we scale up the activation of GRPO reasoning heads with baseline model, up to 1.3 higher activation, we observe actual performance gain with the heads from 100 step checkpoints GRPO GSM8K circuits. The performance of MATH benchmark increases from 56 to 60, while other benchmarks like AIME'24 and AMC decrease. Meanwhile, when we scale up 1.3 higher for the one head emergent from 2500 step checkpoints GRPO Math-220k, the performance of AMC goes slightly down, and MATH goes slightly up, while AIME'24 remains static. On the other side, when we scale down by half using that same attention heads emerging from 100 step checkpoints GRPO GSM8K circuits, AIME'24 performance decreases sharply from 13.3 to 3.3. However, MATH and AMC score increase, 56 to 63, and 38.5 to 42.1. This trade-off is impressive as some task specific heads affect strongly to that performance, while it may harm or make model confused to do other tasks. As heads of 100 step checkpoints GRPO GSM8K circuits are mostly coming from AIME dataset basis, it surely affect AIME the most, while scaling down its presence could help model to do reasoning better at other benchmarks. Detail of score is in the Table 5.

Additional test in Figure 5 shows the trade-off of performance after GRPO. Training with GSM8K dataset has an early sweet spot: at 100–200 steps, AIME and AMC rise from their baselines (13.3 and 38.6) to around 20 and 43, and MATH from 56 to about 67. However, later checkpoints gradually lose these gains with an sign of overfitting, such as rigidly applying only specific solving problem. With OpenR1-Math, AIME becomes highly unstable and fluctuate, where some checkpoints shows both effective solving strategies and stuck in fuction calling loop.

**Qualitative Analysis.** As GRPO sharpens multi-step mathematical reasoning and problem structuring, it yields better reasoning on composite word problems when we qualitatively compare it against baseline model's one. However, it also degrades basic numeracy, execution stability, and tool-choice agility. For the early checkpoints of OpenR1-Math-220k and GSM8K, they show gains in symbolic manipulation with fewer end-stage slips, yet prefer cumbersome analytic derivations over simple programmatic checks. For mid-later checkpoints, which show lower performance than others, they exhibit overfitting and forgetting signs for the core algebra and geometry. Overall, GRPO yields clearer, more systematic reasoning traces and improved strategy formation, but can erode numeracy and robustness when optimization pressure or dataset style dominates. Examples of qualitative analysis are given in the Appendix A.8.2 and A.9.2.

Table 2: Emergent head ablation inference for Qwen3-8B. Every performance is measured with pass@1 score with temperature 0.6 and 32k context length, as Yang et al. (2025) suggested for the best performance setting. Each ablation cases make the value of specific attention heads, around 5 to 10 number of heads from its circuit results, into zero or scale down to half for checking its importance for reasoning tasks. As no other reasoning heads are found among thinking mode, we do ablation only for thinking off mode. We color some scores into red for the most degraded results and green for the most performance improvement. Ablating overstuffed attention heads in thinking off mode increases the baseline score with minimal performance trade-offs.

| Model | Method | AIME'24 | AIME'25 | AMC | GPQA | MATH |
|-------|--------|---------|---------|-----|------|------|
| Qwen3-8B | Think On | 80.0 | 73.3 | 89.1 | 63.1 | 93.8 |
| | Think Off | 30.0 | 13.3 | 67.4 | 44.9 | 81.4 |
| | Think Off & Ablation | 36.6 | 20.0 | 61.4 | 49.4 | 83.6 |
| | Think Off & Scale Down | 20.0 | 23.3 | 56.6 | 51.0 | 81.8 |

**Re-Implementation for Generalizability.** We apply the same analysis to the Llama-3.2 model, with results shown in Figure 12. The overall trend is qualitatively similar to Qwen2.5-Math: attention heads emerge and then disappear across training checkpoints as the model searches for effective reasoning pathways. However, unlike Qwen2.5-Math GRPO, Llama-3.2 GRPO activates a much larger number of heads from checkpoint 500 onward, and these newly active heads are spread across early-to-mid layers rather than concentrating into a few specific positions. We hypothesize that this difference stems from Llama-3.2's weaker base capability: although GRPO is intended to sharpen a small set of reasoning circuits, in this low-capability regime it instead seems to spend capacity on broadly lifting generic skills. This yields a head-usage pattern closer to SFT than to a compact circuit, and we stress that this is a correlational observation rather than a causal claim.

## 6 In-Depth Analysis on Think On/Off

Recently suggested thinking on/off functionality in models provides a unique window into how efficient reasoning is implemented (Tu et al., 2025). Efficiently controlling reasoning level is distinct among architectures, for example, system level routing to select between the fast model and the deeper reasoning model (OpenAI, 2025a), and using system message keyword to control reasoning level (Agarwal et al., 2025). In this work, we implement Qwen3-8B (Yang et al., 2025) as it enable controlled circuit comparison under an instruct-style template with explicit thinking on/off gating using <think> token, yielding clean think on versus off conditions.

**Think off compensate performance through enormous head emerging.** Our analysis suggests that "Think-On" triggered in chat template is not about activating specific, monolithic attention heads, but about selecting the most efficient computational pathways within overall attention head pool. Here, circuits constructed from the default think-on mode are not composing a set of unique, reasoning-only heads. Instead, it relatively shares most of its components with the think-off mode circuits. Interestingly, when the thinking is disabled by predefined <think>\n</think> token within chat template, the model activates much larger number of attention heads. This observation suggests that the model has internalized a highly efficient mechanism for selecting reasoning pathway.

While this differs from phenomena observed in post-training methods like GRPO, where reasoning-specific heads exits, the integrated nature of Qwen3, unifying a general instruction following (think-off mode) with a reasoning capability (think-on mode), appears to have fostered an ability to find the most resource efficient path. When the specialized reasoning pathway is explicitly disabled (think mode is off), the model compensates for it by activating a broader, more redundant network of attention heads. In contrast, the think-on mode allows it to engage a specific, optimized circuit already embedded within its structure, demonstrating an advanced form of learned computational efficiency.

**Result of Head Intervention.** Table 2 shows our quantitative analysis with head intervention for each benchmark performance. We implement the attention head ablation and head activation scale down for those heads found exclusively in think off circuits. Without thinking mode, model's performance drops significantly, especially for hard level benchmarks such as AIMEs. We find that if we ablate parts of think off circuit heads in thinking off mode, the removal of overly activated and confusing attention heads clarifies the model's reasoning pathways, leading to improved performance across multiple benchmarks. The most effective benchmarks are AIME'24 and 25, which

demand more complex and well structured mathematical reasoning compared with other benchmarks. Meanwhile, scaling down the activation of those think off circuit heads in half also contributes to the performance gain, even higher than ablation in some benchmarks like GPQA and AIME'25. It also results in some trade off as the score of AIME'24 decreases from 30 to 20.

**Performance Difference Against Coverage Comparison.** To further investigate performance under varying sampling coverage, we compare the models' pass@k scores on AIME'24 with up to 64 samples. Detail of metric is in §A.7. As shown in Figure 6 (left), the baseline think-off model consistently maintains a slight performance advantage as $k$ increases. We hypothesize that its large number of active attention heads facilitates the exploration of diverse reasoning pathways, a benefit that scales with the number of samples. In contrast, the ablation and scale downed models exhibit a diminished capacity to discover novel solutions at higher $k$ values and large $n$ samples. This behavior is reminiscent of models that, after post-training like GRPO, become locked into specific reasoning paths and fail to solve certain problems regardless of the increased coverage (Yue et al., 2025).

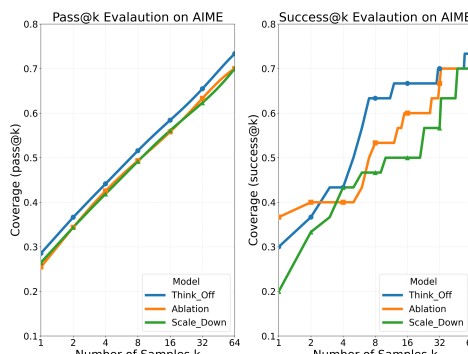

Figure 6: Performance difference against increasing coverage. The left figure shows pass@k difference when sampling coverage increased, while the right figure shows efficient correctness with success@k.

This trade-off is more starkly illustrated when analyzing generation efficiency, success@k, which calculates the probability of finding a correct solution within each trial, as shown in Figure 6 (right). Here, the ablation model initially outperforms the baseline at very low sampling rates ($k \leq 2$), suggesting that simplification of attention heads helps focus the model on a more direct and efficient reasoning path. However, this advantage quickly vanishes as $k$ increases, where the baseline's ability to explore a wider solution space becomes more fruitful. Meanwhile, the scale down model consistently under-performs, appearing to lack both the focused efficiency of the ablated model and the exploratory breadth of the baseline. Collectively, these results highlight the dual nature of the numerous emergent heads in the think off mode: they can introduce noise in low-sample scenarios but become a crucial asset for robust problem-solving when a larger computational budget is available.

## 7 RELATED WORK

### 7.1 SUPERVISED FINE-TUNING (SFT) & DISTILLATION

Post-training is a crucial stage that adapts a general-purpose pretrained LLM for specialized tasks such as complex reasoning (Zhang et al., 2025a). Supervised Fine-Tuning (SFT) adapts a pretrained model to a specific tasks by training it on a curated dataset of input-output examples (Wei et al., 2022a). In the context of reasoning, a powerful technique is to use a large, more capable "teacher" model (e.g., DeepSeek-R1 (Guo et al., 2025)) to generate high-quality, step-by-step reasoning races, often called Chain-of-Tought (CoT) (Wei et al., 2022b) prompts. A smaller "student" model is then fine-tuned on this synthetic dataset, learning to mimic the teacher's reasoning process (Kang et al., 2023). SFT forces the student model's output distribution to match the teacher's, and this direct and forceful adaptation often results in significant, dense updates to the model's parameter by memorizing specific reasoning paths (Chu et al., 2025). This form of knowledge distillation has proven effective for creating capable open-source reasoning models (Toshniwal et al., 2024). In this work, we utilize distilled version of DeepSeek-R1 for the corresponding Qwen2.5 Math (Yang et al., 2024), and do SFT with sampled OpenR1-Math-220k dataset for comparison (Hugging Face, 2025).

### 7.2 REINFORCEMENT LEARNING WITH VERIFIABLE REWARDS (RLVR)

Reinforcement learning (RL) offers an alternative paradigm where a model learns by interacting with an environment and receiving reward signals (Ouyang et al., 2022). It is particularly well-suited for tasks like the mathematical reasoning where the correctness of a final answer can be automatically verified, providing a clear, albeit sparse, reward signal. This Reinforcement Learning with Verifiable Rewards (RLVR) allows the model to explore different reasoning paths and reinforces those that lead to correct outcomes, without being constrained to a signal gold path as in SFT. A prominent RL

algorithm used for training reasoning models is Group Relative Policy Optimization (GRPO) (Shao et al., 2024), a variant of Proximal Policy Optimization (PPO) (Schulman et al., 2017), designed to be more memory efficient and stable training. We adopt GRPO to implement RLVR for mathematical reasoning; the full objective and training formulation are detailed in §A.6.

### 7.3 MECHANISTIC INTERPRETABILITY

Mechanistic interpretability seeks to explain model behavior via internal mechanisms, and one common approach studies small, causally meaningful "circuits" connecting attention heads and MLPs (Nanda et al., 2023; Conmy et al., 2023; Ameisen et al., 2025; Lindsey et al., 2025). Circuits have been reverse-engineered for indirect object identification in GPT-2 Small (Wang et al., 2023), for factual and temporal knowledge (Yao et al., 2024b; Park et al., 2025), and for chain-of-thought reasoning (Dutta et al.; Cabannes et al., 2024), while arithmetic work shows that models rely on a "bag of heuristics" implemented by sparse MLP features rather than a single clean algorithm (Nikankin et al.). However, interpreting such circuits at the level of individual units is complicated by *polysemanticity*: superposition makes neurons and heads mix multiple unrelated features, and many human-interpretable features appear only as sparse combinations of neurons rather than clean single units (Elhage et al., 2022; Scherlis et al., 2022; Gurnee et al.).

This has motivated feature-based approaches such as Sparse Feature Circuits and their large-scale extensions (Marks et al.; Caples et al., 2025) and Transcoder-based MLP replacements (Dunefsky et al.), which learn sparse latent features for more precise circuit editing but require substantial extra training and are currently implemented for only a few architectures. Head- and neuron-level circuit analyses nonetheless remain the default abstraction in transformer-circuits work and continue to yield experimentally testable insights (Wang et al., 2023; Yao et al., 2024b; Park et al., 2025), so we adopt this conventional perspective and operate directly on native attention heads. By avoiding per-layer sparse autoencoders or transcoders, our analysis is much more computationally efficient and easily transferable across architectures and post-training regimes, at the cost of some residual polysemanticity. Most closely related to our goals, (Prakash et al., 2024) find that fine-tuning on entity tracking mainly strengthens existing mechanisms rather than creating new ones, whereas in our math-only SFT and GRPO setting with an explicit `<think>` token we observe emergent "reasoning heads" that are negligible in the base model but become critical after post-training, suggesting that circuit reorganisation depends strongly on both task domain and training paradigm.

## 8 CONCLUSION, LIMITATION, FUTURE WORK

We present comparative, mechanistic account of how post-training paradigms reconfigure the internal mechanism of reasoning models. Our analyses show that these methods do not merely explore a fixed parameter landscape, instead, they reshape functional structure: distillation and SFT steadily embed new computational pathways via the sustained emergence of additional, large reasoning heads, on the other hand, GRPO conducts reward-guided head configurations, with heads appearing and being pruned over training, to optimize capabilities. The think on/off architecture behaves as a selective gate, as thinking mode activates just the task-relevant heads, while thinking off compensates ability through more diverse attentions with enormous heads. And their differences align with observed performance trade-offs: the systems more often solve hard problems by forming deeper, more structured plans, yet sometimes regress on previously easy items due to over reasoning or arithmetic slips.

Although this provides a new lens through which to view post-training, our findings are constrained by two factors. First, the generalizability of our implementation has only been validated on the Qwen model series and single Llama model. Although our re-implementation on Llama confirms a relatively effective transition, further work is necessary to establish its effectiveness across a broader spectrum of model architectures. Second, our analysis relies on prompt-based circuits, which demand precise setup and may be vulnerable to polysemanticity. While alternative approaches like SAE-based circuits could mitigate this issue, we deemed them impractical for this study, as they are computationally costly and less generalizable, requiring separate SAEs to be trained for every checkpoints.

Still, its conclusions are subject to offer avenues for future research. Taken together, our results motivate attention head informed training policies that (i) encourage targeted head activation rather than uncontrolled head growth, (ii) use reward shaping to jointly optimize plan quality and calculation reliability, and (iii) leverage per-head influence estimates to guide selective post-training. We view this mechanistic perspective as a foundation for principled, interpretable, and robust post-training of effective reasoning strategies with the assurance of reliable, flawless execution.

DECLARATION ON GENERATIVE AI

During the preparation of this work, the author(s) used Gemini 2.5 Pro in order to: Grammar, spelling check and latex format check.

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

# A APPENDIX

## A.1 EXPERIMENTAL SETUP

**Models.**  We select a consistent family of models to serve as the testbed for our analysis among similar architecture and design. As Qwen series make it possible to compare almost every possible reasoning training, we specifically pick this model variations and analyze deeply. The models include:

- Baseline Models: Qwen2.5-Math-1.5B-Instruct and Qwen2.5-Math-7B-Instruct (Yang et al., 2024), which are strong base models pretrained with a focus on mathematical capabilities.
- Distilled Models: DeepSeek-R1-Distill-Qwen-1.5B and 7B (Guo et al., 2025), which represent the outcome of knowledge distillation from a powerful teacher reasoning model.
- Think On/Off Model: Qwen3-8B (Yang et al., 2025), which features a Think On/Off capability across various open source models, allowing for controlled study of selective reasoning activation.

We additionally adopt Llama-3.2-1B-Instruct (Meta, 2024b) for generalizable re-implementation, though it cannot be compared with the corresponding DeepSeek distillation and think on/off model as they do not exist.

**Datasets.**  Our training and evaluation cover the well-established, widely-used reasoning datasets:

- Training: For SFT and GRPO, we utilize standard, large-scale reasoning datasets, including OpenR1-Math-220k (Hugging Face, 2025) and GSM8K (Cobbe et al., 2021), which contain a diverse set of mathematical problems and their solutions.
- Evaluation: To assess both in-domain and out-of-domain generalization, we employed a comprehensive suite of benchmarks: AIME'24 and AIME'25 (American Invitational Mathematics Examination) (AIME, 2025), AMC (American Mathematics Competitions) (AI-MO, 2024), GPQA (Graduate-Level Google-Proof Q&A) (Rein et al., 2024), MATH-500 (Lightman et al., 2024) and TriviaQA (Joshi et al., 2017) for general knowledge.

**Training & Evaluation.**  For each post-training method, we follow established best practices and maintain consistent hyperparameters where possible to facilitate fair comparison. For GRPO, we train a Qwen2.5-Math-1.5B-Instruct for 3 epochs, saving checkpoints every 100 steps to enable a temporal analysis of circuit formation. For SFT, we used a setup designed to mirror the GRPO training process in terms of data exposure. We also utilize Light-R1 (Wen et al., 2025) as our codebase, modifying it so that the pass@1 evaluation metric is computed as the average over multiple responses for each setting. All training and inference are done with two NVIDIA H100 GPUs(80GB). Hyper-parameter setup for each post-training is like below:

- SFT (Wei et al., 2022a): learning rate $4.0e-5$, 5 training epochs, 100 steps for saving and circuit construction, Bfloat16, warm-up ratio 0.03. For Llama3.2 1B: learning rate $4.0e-5$, 5 training epochs, 100 steps for circuit construction, Bfloat16, warm-up ratio 0.03
- GRPO (Shao et al., 2024) with OpenrR1-Math-220k: learning rate $1.0e-6$ for main result and $2.0e-5$ for comparison in Figure 8, 3 training epochs, 100 steps for saving and circuit construction, Bfloat16, warm-up ratio 0.1, reward_weights 1.0, 16 generations. For Llama3.2 1B: learning rate $2.0e-7$, 3 training epochs, 100 steps for saving and circuit construction, Bfloat16, warm-up ratio 0.1, reward_weights 1.0, 16 generations.
- GRPO (Shao et al., 2024) with GSM8K: learning rate $5e-6$, 1 training epoch, 100 steps for saving and circuit construction, Bfloat16, warm-up ratio 0.1, reward_weights 1.0, 16 generations.

For the system prompt of GRPO training, we use basic recipes of OpenR1 Hugging Face (2025).

> You are a helpful AI Assistant that provides well-reasoned and detailed responses. You first think about the reasoning process as an internal monologue and then provide the user with the answer. Respond in the following format:
> `<think>\n...\n</think>\n<answer>\n...\n</answer>`

## A.2 CIRCUIT CONSTRUCTION SETUP

We construct circuits using EAP-IG (Hanna et al., 2024), where *ig-step* is 100 and *top-n* is 5000. We also simplify each circuits with the threshold $\tau = 0.1$ for filtering out important edges and nodes. Examples of simplified circuits among various models are in Figure 16, 17, and 18. Figure 19 is the examples of simplified circuits with Llama3.2 1B.

**Prompt Settings.** We sample various responses of baseline models and reasoning models, then make an input prompt for circuit construction using chat template.

---
**Reasoning Model**

<think>Okay, so I have this problem where Aya goes ...

- - - - - - - - - - - - - - - - - - - - - - - - - - - - - - - - - - - - - - - - - - - - - - - - - - -

<think>Alright, so I have this geometry problem here ...

- - - - - - - - - - - - - - - - - - - - - - - - - - - - - - - - - - - - - - - - - - - - - - - - - - -

<think>Okay, so I need to find the eigenvector ...

- - - - - - - - - - - - - - - - - - - - - - - - - - - - - - - - - - - - - - - - - - - - - - - - - - -

<think>...

---
**Baseline model**

We'll use Python to help us ...

- - - - - - - - - - - - - - - - - - - - - - - - - - - - - - - - - - - - - - - - - - - - - - - - - - -

To determine the molecular

---

For Llama3.2 1B, we sample responses of baseline models and after reasoning to construct circuits.

---
**Models After Post Training**

<think>Step 1: Define the variables and given conditions Let's denote ...

---

---
**Models Before Post Training**

To solve this problem, we'll ...

---

## A.3 DETAIL OF EAP-IG CALCULATION

**Global path.** The IG path is defined over the *entire token-embedding sequence*: we linearly interpolate between corrupted and clean inputs as $z' + \alpha(z - z')$ with $\alpha = \frac{k}{m}$, $k = 1, \ldots, m$. No pooling into a single "document embedding" is used.

**Input of $v$.** For a node $v$ (attention head block or MLP), the "input of $v$" is the *residual-stream pre-activation* that $v$ receives at its destination positions, i.e., the sum of all parents' outputs just before $v$ applies its operation. Accordingly, the gradient in equation 3 is $\nabla_{z_v}\mathcal{L}$ with respect to that residual vector.

**Token granularity and per-example score.** While the path lives in sequence space, the edge score for $(u \to v)$ is evaluated at coordinates corresponding to $(v)$'s destination positions. For next-token objectives we use the position $(t)$ whose logits are evaluated; for sequence-level objectives we average over supervised positions $(T^*)$. The per-example score is

$$\text{score}(u \to v \mid x) = \left\langle \Delta z_u(x), \; \frac{1}{m}\sum_{k=1}^{m} \left(\nabla_{z_v}\mathcal{L}\right)\Big|_{z' + \frac{k}{m}(z - z')} \right\rangle, \tag{4}$$

where $\langle \cdot, \cdot \rangle$ denotes the dot product in the residual dimension.

**Aggregation and selection.** We rank edges using a dataset aggregate, e.g., $\mathbb{E}_x[|\text{score}(u \to v \mid x)|]$. Using $\Delta z_u = z_u - z'_u$ or $z'_u - z_u$ only flips the sign; absolute aggregation makes ranking invariant. We select top-$n$ edges, prune isolated nodes, and validate faithfulness by ablating all non-circuit edges.

**Practical choices.** We typically use $m \in [5, 8]$ Riemann steps and a task-agnostic divergence (e.g., KL) computed at the same evaluation positions as above; rankings are robust without extra normalization, though optional rescaling can be applied for cross-model comparability.

## A.4 DETAIL OF EFFECT AND IMPORTANCE MEASURE

Our effect analysis reuses the EAP-IG edge scores already computed for circuit extraction (§ 2). For a given model $M$ and input $x$ from a benchmark dataset $\mathcal{D}$, EAP-IG assigns to each edge $(u \to v)$ in the circuit $C^{(M)}(x)$ a scalar attribution score $s_x^{(M)}(u \to v) \in \mathbb{R}$, which we obtain after thresholding on $|s_x^{(M)}(u \to v)|$ to keep only top-attribution edges. We treat attention heads as modules and aggregate edge-level scores into a head-level importance matrix.

**Head-level importance.** Let $a_{\ell,h}$ denote the attention head at layer $\ell$ and index $h$. For model $M$, we define the (unnormalized) importance of $a_{\ell,h}$ as the sum of absolute EAP-IG scores over all circuits and all edges whose source node is that head:

$$\tilde{I}_M(\ell, h) = \sum_{x \in \mathcal{D}} \sum_{\substack{(u \to v) \in C^{(M)}(x) \\ u = a_{\ell,h}}} \left| s_x^{(M)}(u \to v) \right|. \tag{5}$$

To allow comparison across models, we apply a global normalization so that the total mass of importance is 1:

$$I_M(\ell, h) = \frac{\tilde{I}_M(\ell, h)}{\sum_{\ell', h'} \tilde{I}_M(\ell', h')}. \tag{6}$$

This yields a head-level importance matrix $I_M \in \mathbb{R}_{\geq 0}^{L \times H}$, where $L$ is the number of layers and $H$ the number of heads per layer.

**Effect measure between pre- and post-trained models.** Given a pre-trained (base) model $M_{\text{pre}}$ and a post-trained model $M_{\text{post}}$ (e.g., DeepSeek-distilled, SFT, or GRPO-trained), both evaluated on the same dataset $\mathcal{D}$ with identical EAP-IG hyperparameters and edge-thresholding, we quantify the change in importance of head $(\ell, h)$ by the symmetric effect measure

$$E(\ell, h) = \frac{I_{M_{\text{post}}}(\ell, h) - I_{M_{\text{pre}}}(\ell, h)}{I_{M_{\text{post}}}(\ell, h) + I_{M_{\text{pre}}}(\ell, h) + \varepsilon}, \tag{7}$$

where $\varepsilon > 0$ is a small constant (we use $\varepsilon = 10^{-6}$) to avoid division by zero. By construction, $E(\ell, h) \in [-1, 1]$, with positive values indicating increased attribution-based importance of $a_{\ell,h}$ in the post-trained model and negative values indicating decreased importance.

For training regimes with multiple checkpoints $M_{\text{post}}^{(t)}\}_{t \in \mathcal{T}}$ (e.g., SFT or GRPO), we compute equation 7 for each checkpoint $t$ to obtain $E^{(t)}(\ell, h)$ and then aggregate along the time axis via a simple arithmetic mean:

$$\bar{E}(\ell, h) = \frac{1}{|\mathcal{T}|} \sum_{t \in \mathcal{T}} E^{(t)}(\ell, h). \tag{8}$$

The resulting matrix $\bar{E} \in [-1, 1]^{L \times H}$ is visualized as the effect heatmaps in Figure 13, where blue (red) cells correspond to heads whose EAP-IG importance increases (decreases) relative to the base model. Note that, because circuits are defined using a fixed attribution threshold, these measures capture importance reallocation within the *top-attribution circuits* considered in our analysis.

## A.5 DETAIL OF SPARSE FEATURE CIRCUIT ANALYSIS

**Construction of Graph.** Constructing full Sparse Feature Circuits (Marks et al.) implies a prohibitive computational cost, scaling with the number of training methods, model checkpoints, layers, and components. To make this tractable while leveraging the disentanglement benefits of Sparse Autoencoders (SAEs) (Bricken et al., 2023), we limit our scope to a direct comparison between Llama-3.1-8B (Base) (Meta, 2024a) and DeepSeek-R1-Distill-Llama-8B (Guo et al., 2025), where both model's full SAEs for residual stream are available through Neuronpedia (He et al., 2024). We utilize those pre-trained Residual Stream SAEs to decompose residual activations into sparse features $f \in \mathbb{R}^{d_{\mathrm{SAE}}}$. However, for Attention and MLP blocks where SAE training is computationally demanding, we retain a dense representation using *identity dictionaries*, and compute attribution scores for sparse features in the residual stream and for dense block outputs in the Attention and MLP layers using the same mathematical algorithm as in EAP-IG (Hanna et al., 2024). Input dataset is same with previous EAP-IG analysis, which is AIME base prompt with sampled answer.

**Aggregated Importance and Shift Measurement.** Since the learned dictionary bases of SAEs differ between the base and post-trained models, a direct feature-to-feature comparison is infeasible. Instead, we aggregate importance at the component level to quantify macroscopic shifts. For a model $M$, layer $\ell$, and component $c \in \{\mathrm{RESID}, \mathrm{ATTN}, \mathrm{MLP}\}$, the importance $I_M(\ell, c)$ is the sum of absolute attribution scores of all constituent nodes (active SAE features for Resid, or the dense block for Attn/Mlp). We then visualize the shift using the symmetric relative difference defined in §A.4:

$$E(\ell, c) = \frac{\hat{I}_{M_{\mathrm{post}}}(\ell, c) - \hat{I}_{M_{\mathrm{pre}}}(\ell, c)}{\hat{I}_{M_{\mathrm{post}}}(\ell, c) + \hat{I}_{M_{\mathrm{pre}}}(\ell, c) + \varepsilon}, \tag{9}$$

where $\hat{I}$ denotes the globally normalized importance. This metric highlights which computational stages become more critical after distillation.

**Results and Discussion.** The analysis reveals distinct patterns in computational reallocation. Figure 14 shows component-level importance with a single heatmap. Consistent with our head-level EAP-IG findings, we observe a strong emergence of importance in **Layer 0 Attention**, suggesting early-stage emergence of attention heads remains crucial. Notably, the **Residual Stream** features exhibit a progressive strengthening in the mid-to-late layers, indicating a reliance on deep, disentangled representations for reasoning. The **MLP** blocks also show increased importance in later layers, albeit less dominantly than residuals. While this SAE-based approach offers reduced polysemanticity and corroborates our main findings, its coarse granularity at the Attention/MLP block level prevents the precise identification of specialized heads. Therefore, given the trade-off between feature interpretability from enormous computational cost and practical granular component tracking, we retain the standard head-level EAP-IG as our primary analytical framework.

## A.6 DETAIL OF GRPO FORMULATION

For a prompt $q$, sample $G$ candidate responses $\{o_i\}_{i=1}^{G}$ from the old policy $\pi_{\mathrm{old}}$; the policy parameters $\theta$ are updated to maximize

$$\mathcal{J}_{\mathrm{GRPO}}(\theta) = \mathbb{E}\left[\frac{1}{G}\sum_{i=1}^{G}\frac{1}{|o_i|}\sum_{t=1}^{|o_i|}\min\Big(r_{i,t}(\theta)\,\hat{A}_{i,t},\ \mathrm{clip}\big(r_{i,t}(\theta),\, 1-\epsilon,\, 1+\epsilon\big)\,\hat{A}_{i,t}\Big)\ -\ \beta\,D_{\mathrm{KL}}\big(\pi_\theta\,\|\,\pi_{\mathrm{ref}}\big)\right], \tag{10}$$

where the token-level policy ratio is

$$r_{i,t}(\theta) = \frac{\pi_\theta(o_{i,t}\mid q, o_{i,<t})}{\pi_{\mathrm{old}}(o_{i,t}\mid q, o_{i,<t})}. \tag{11}$$

In the outcome-reward variant used for verifiable tasks, a reward model assigns a scalar $R_i$ to each output $o_i$. GRPO then uses a value-free, group-normalized advantage shared across all tokens of $o_i$:

$$\hat{A}_{i,t} = \frac{R_i - \mathrm{mean}(R)}{\mathrm{std}(R)} \quad \text{for all } t \in \{1, \ldots, |o_i|\}, \tag{12}$$

which compares each response to its group peers and obviates a learned critic. The min–clip structure conservatively bounds updates, while the KL regularizer with coefficient $\beta$ constrains divergence from a reference policy $\pi_{\mathrm{ref}}$, improving stability and mitigating reward over-optimization. We specifically implement OpenR1 with the same Math-220k for GRPO training to compare base model with reasoning trained version (Hugging Face, 2025).

## A.7 DETAIL OF EVALUATION

**Generation and Sampling Setup** For our quantitative evaluation, we generate various responses $n = 4$ to $64$ for each problem in the respective test sets. The generation process for each models uses a sampling temperature of $T = 0.6$ and a top-p (nucleus sampling) value of $0.95$, or if the model's best practice is suggested such as Qwen3-8B, we follow those settings; $T = 0.6$, top-p=$0.95$, top-k=$20$, and min-p=$0$ for thinking mode.

**Pass@k for Overall Capability** To assess the overall problem-solving capability of each model, we employ the standard **pass@k** metric, as introduced by Chen et al. (2021). This metric provides an unbiased estimator for the probability that at least one correct solution is generated in $k$ attempts. Given $n$ total generated samples for a problem and $c$ correct samples among them, the pass@k score for that single problem is calculated as:

$$\text{pass@k} = 1 - \frac{\binom{n-c}{k}}{\binom{n}{k}} \tag{13}$$

The final reported pass@k score is the average of these values across all problems in the test set. This metric is independent of the generation order and measures the model's theoretical potential to solve a problem given a budget of $k$ samples.

**Success@k for Generation Efficiency** While pass@k measures overall capability, it is agnostic to the generation order. To measure the practical generation efficiency, a model's ability to find a correct solution quickly, we also compute **success@k**. This metric evaluates the likelihood of finding a correct solution within the trial $k$ sequentially generated samples.

Let $R_p = (r_1, r_2, \ldots, r_n)$ be the ordered sequence of responses for a problem $p$, and let $v(r_i)$ be a verification function that returns 1 if response $r_i$ is correct and 0 otherwise. The success@k is then the average success rate across all problems:

$$\text{success@k} = \mathbb{E}_{\text{problems}} \left[ \mathbb{1}\left\{ \sum_{i=1}^{k} v(r_i) > 0 \right\} \right] \tag{14}$$

where $\mathbb{1}\{\cdot\}$ is the indicator function. This metric directly rewards models that produce correct answers earlier in the generation process. The comparison between pass@k and success@k allows us to disentangle a model's latent capability from its practical efficiency, as discussed in our main analysis.

## A.8 EXAMPLE OF QUALITATIVE ANALYSIS: NEWLY CORRECT

---

**Input Question from AIME'24**

Every morning Aya goes for a 9-kilometer-long walk and stops at a coffee shop afterwards. When she walks at a constant speed of $s$ kilometers per hour, the walk takes her 4 hours, including $t$ minutes spent in the coffee shop. When she walks $s + 2$ kilometers per hour, the walk takes her 2 hours and 24 minutes, including $t$ minutes spent in the coffee shop. Suppose Aya walks at $s + \frac{1}{2}$ kilometers per hour. Find the number of minutes the walk takes her, including the $t$ minutes spent in the coffee shop. Let's think step by step and output the final answer within \boxed{}.

---

Baseline model's response just repeat same equation endlessly without specific conclusions or reasoning. We qualitatively evaluate how each post-training's answer is different from the original.

---

**Response of Baseline model (Qwen2.5-Math-1.5B)**

To solve this problem, we need to establish the relationship between the walking speed and the time taken for the walk, including the time spent in the coffee shop. Let's break down the problem step by step. 1. **Define Variables:** - Let $s$ be the walking speed in kilometers per hour. - Let $t$ be the time spent in the coffee shop in hours. - The total time for the walk, including the coffee shop time, is given by $\frac{9}{s} = 4$ hours and $\frac{9}{s+2} = 2.4$ hours. 2. **Set Up Equations:** - From the first condition: $\frac{9}{s} = 4$ which gives $s = \frac{9}{4}$. - From the second condition: $\frac{9}{s+2} = 2.4$ which gives $s + 2 = \frac{9}{2.4} = \frac{90}{24} = \frac{15}{4}$ and thus $s = \frac{15}{4} - 2 = \frac{15}{4} - \frac{8}{4} = \frac{7}{4}$. Since we have two different values for $s$ from the two conditions, we need to verify which one is correct. Let's check both conditions with $s = \frac{9}{4}$ and $s = \frac{7}{4}$. 3. **Verify the Values:** - For $s = \frac{9}{4}$: - Time for the walk: $\frac{9}{\frac{9}{4}} = 4$ hours. - Time for the walk at $s + 2 = \frac{17}{4}$: $\frac{9}{\frac{17}{4}} = \frac{36}{17} \approx 2.12$ hours, which is not 2.4 hours. - For $s = \frac{7}{4}$: - Time for the walk: $\frac{9}{\frac{7}{4}} = \frac{36}{7} \approx 5.14$ hours, which is not 4 hours. ... (endless repeat...)

---

### A.8.1 SFT

---

**Response of SFT checkpoint 1000**

`<think>` Okay, let's try to figure out this problem. So, Aya walks 9 kilometers every morning, and her walking speed is either s km/h or s+2 km/h. When she walks at s km/h, the total time, including a coffee shop break of t minutes, is 4 hours. When she walks faster at s+2 km/h, the total time is 2 hours and 24 minutes, still including the same t minutes in the coffee shop. We need to find the total time when she walks at s+0.5 km/h, including the coffee shop time. First, let's parse the problem. The key information is: 1. Distance: 9 km both times. 2. Walking speed: s km/h initially, then s+2 km/h. 3. Time taken: 4 hours at speed s, 2 hours 24 minutes at speed s+2, with both include t minutes in the coffee shop. 4. Find the total time when walking at s+0.5 km/h, including t minutes. So, the total time is the walking time plus the coffee shop break. The walking time can be calculated by distance divided by speed. The coffee shop break is t minutes each time, regardless of speed. Let me write down the equations. First scenario: walking speed s km/h, total time 4 hours, including t minutes. So the walking time is 4 hours minus t minutes. Wait, but time is usually in hours in these problems. So maybe we need to convert everything to hours. ... But according to the problem, when walking at 4.5 km/h, the total time is 2.4 hours, which is 144 minutes, which is correct. So that checks out. Therefore, the total time when walking at 3 km/h is 3 hours 24 minutes, which is 3*60 +24=204 minutes. So the answer is 204 minutes, so \boxed{204} **Final Answer** \boxed{204} `</think>` Given that Aya walks 9 kilometers every morning, ...

---

### A.8.2 GRPO

---

**Response of GRPO checkpoint 2500**

Please reason step by step, and put your final answer within \boxed{}. Offset OID Every morning Aya goes for a 9-kilometer-long walk and stops at a coffee shop afterwards. When she walks at a constant speed of $s$ kilometers per hour, the walk takes her 4 hours, including $t$ minutes spent in the coffee shop. When she walks $s + 2$ kilometers per hour, the walk takes her 2 hours and 24 minutes, including $t$ minutes spent in the coffee shop. Suppose Aya walks at $s + \frac{1}{2}$ kilometers per hour. Find the number of minutes the walk takes her, including the $t$ minutes spent in the coffee shop. First, let's establish the relationships from the given information. 1. When walking at $s$ kilometers per hour, the total time is 4 hours, including $t$ minutes in the coffee shop. Since 4 hours is 240 minutes, we can write:

$$\text{Time spent walking} = \frac{9}{s} \text{ hours} = \frac{9}{s} \times 60 \text{ minutes} = \frac{540}{s} \text{ minutes.}$$

So, the total time including the coffee shop time is:

$$\frac{540}{s} + t = 240.$$

This simplifies to:

$$\frac{540}{s} = 240 - t. \tag{1}$$

2. When walking at $s + 2$ kilometers per hour, the total time is 2 hours and 24 minutes, which is 144 minutes. So, we can write:

$$\text{Time spent walking} = \frac{9}{s + 2} \text{ hours} = \frac{9}{s + 2} \times 60 \text{ minutes} = \frac{540}{s + 2} \text{ minutes.}$$

So, the total time including the coffee shop time is:

$$\frac{540}{s + 2} + t = 144.$$

... Now, we need to find the time it takes Aya to walk at $s + \frac{1}{2} = 2.5 + 0.5 = 3$ kilometers per hour:

$$\text{Time spent walking} = \frac{9}{3} \text{ hours} = \frac{9}{3} \times 60 \text{ minutes} = 180 \text{ minutes.}$$

Including the 24 minutes in the coffee shop, the total time is:

$$180 + 24 = 204 \text{ minutes.}$$

The final answer is: \boxed{204}. ...

---

## A.9 EXAMPLE OF QUALITATIVE ANALYSIS: NEWLY INCORRECT

---

**Input Question from AIME'24**

Find the largest possible real part of

$$(75 + 117i)z + \frac{96 + 144i}{z}$$

where $z$ is a complex number with $|z| = 4$. Let's think step by step and output the final answer within \boxed{}.

---

Baseline model correctly calculate the result using python codes in this case.

---

**Response of Baseline model (Qwen2.5-Math-1.5B)**

To find the largest possible real part of $(75 + 117i)z + \frac{96+144i}{z}$ where $z$ is a complex number with $|z| = 4$, we can use the following approach: 1. Let $z = 4e^{i\theta}$ where $\theta$ is a real number. This is because $|z| = 4$. 2. Substitute $z = 4e^{i\theta}$ into the expression $(75 + 117i)z + \frac{96+144i}{z}$. 3. Simplify the expression and find the real part. 4. Maximize the real part with respect to $\theta$. Let's proceed with the calculations: 1. Let $z = 4e^{i\theta} = 4(\cos\theta + i\sin\theta)$. 2. Substitute $z$ into the expression:

$$(75 + 117i)z + \frac{96 + 144i}{z} = (75 + 117i)(4\cos\theta + 4i\sin\theta) + \frac{96 + 144i}{4\cos\theta + 4i\sin\theta}.$$

3. Simplify the expression: ... Thus, the largest possible real part is $540$. Let's confirm this with Python code. '''python import sympy as sp # Define the variables theta = sp.symbols('theta', real=True) z = 4 * (sp.cos(theta) + sp.I * sp.sin(theta)) # Define the expression expr = (75 + 117 * sp.I) * z + (96 + 144 * sp.I) / z # Simplify the expression expr_simplified = sp.simplify(expr) # Extract the real part real_part = sp.re(expr_simplified) # Find the maximum value of the real part max_real_part = sp.maximum(real_part, theta) print(max_real_part) ''' '''output 540 ''' The largest possible real part of $(75+117i)z+\frac{96+144i}{z}$ where $z$ is a complex number with $|z| = 4$ is \boxed{540}.

---

### A.9.1 SFT

At first, it successfully computes the answer with a simplified mathematical expression. However, it continues its computation, then goes wrong with its calculation, even incomplete output format.

---

**Response of SFT checkpoint 1000**

<think> Okay, so I need to find the largest possible real part of the expression

$$(75 + 117i)z + \frac{96 + 144i}{z},$$

where $z$ is a complex number with $|z| = 4$. Hmm, complex numbers and optimization... Let me start by understanding the problem better. First, $z$ is a complex number with magnitude 4. That means $z$ can be written in the form $z = 4e^{i\theta}$, where $\theta$ is a real number between $0$ and $2\pi$. Since $z$ is a complex number on a circle of radius 4, maybe I can parametrize $z$ as $4(\cos\theta + i\sin\theta)$. Then, I can express the given expression in terms of $\theta$ and find its real part. Then, maximize that real part with respect to $\theta$.

Let me write that out. Let $z = 4e^{i\theta}$. Then, the expression becomes:

$$(75 + 117i)(4e^{i\theta}) + \frac{96 + 144i}{4e^{i\theta}}.$$

Simplify each term: ... Therefore, we need to maximize $324\cos\theta - 432\sin\theta$. Hmm, okay. So, the problem reduces to maximizing a linear combination of $\sin$ and $\cos$. There's a standard method for maximizing such expressions.

The expression $a\cos\theta + b\sin\theta$ can be written as $R\cos(\theta - \varphi)$, where $R = \sqrt{a^2 + b^2}$. Then, the maximum value is $R$. So for $324\cos\theta - 432\sin\theta$ (i.e., $a = 324$, $b = -432$), the maximum value is

$$\sqrt{324^2 + (-432)^2} = \sqrt{324^2 + 432^2}.$$

Compute:

$$324^2 = 104{,}976, \qquad 432^2 = 186{,}624, \qquad 324^2 + 432^2 = 291{,}600,$$

so

$$\sqrt{291{,}600} = 540.$$

Therefore, the largest possible real part is 540.

But wait, hold on. Let me check my calculation again because 324 squared is 104,976, 432 squared is 186,624. ... 286

---

### A.9.2 GRPO

Although GRPO gives an ability of complex mathematical reasoning, model's response simply goes wrong with calculation mistakes.

---

**Response of GRPO checkpoint 2500**

Please reason step by step, and put your final answer within \boxed{}.ageration agerationuser
Find the largest possible real part of $(75 + 117i)z + \frac{96+144i}{z}$ where $z$ is a complex number
with $|z| = 4$. To solve this problem, we will use the following steps: 1. Represent the complex
number $z$ in terms of its real and imaginary parts. 2. Express the given expression in terms of
$z$ and its conjugate. 3. Use the constraint $|z| = 4$ to simplify the expression. 4. Maximize
the real part of the expression. Let's start by representing $z$ as $z = 4e^{i\theta} = 4(\cos\theta + i\sin\theta)$.
Given the expression $(75 + 117i)z + \frac{96+144i}{z}$, we substitute $z = 4e^{i\theta}$:

$$(75 + 117i)z = (75 + 117i) \cdot 4e^{i\theta} = 4(75 + 117i)(\cos\theta + i\sin\theta)$$

$$\frac{96 + 144i}{z} = \frac{96 + 144i}{4e^{i\theta}} = \frac{96 + 144i}{4}e^{-i\theta} = (24 + 36i)(\cos\theta - i\sin\theta)$$

So the expression becomes:

$$4(75\cos\theta + 117\sin\theta + i(75\sin\theta - 117\cos\theta)) + (24\cos\theta + 36\sin\theta - i(24\sin\theta - 36\cos\theta))$$

Combining the real and imaginary parts, we get:

$$(4 \cdot 75\cos\theta + 4 \cdot 117\sin\theta + 24\cos\theta + 36\sin\theta) + i(4 \cdot 75\sin\theta - 4 \cdot 117\cos\theta - 24\sin\theta + 36\cos\theta)$$

$$= (300\cos\theta + 468\sin\theta + 24\cos\theta + 36\sin\theta) + i(300\sin\theta - 468\cos\theta - 24\sin\theta + 36\cos\theta)$$

$$= (324\cos\theta + 504\sin\theta) + i(276\sin\theta - 432\cos\theta)$$

The real part of this expression is $324\cos\theta + 504\sin\theta$. To maximize this, we use the fact
that the maximum value of $a\cos\theta + b\sin\theta$ is $\sqrt{a^2 + b^2}$. Here, $a = 324$ and $b = 504$, so the
maximum value is:

$$\sqrt{324^2 + 504^2} = \sqrt{(324 \cdot 3) \cdot (1764 + 252)} = \sqrt{324^2 + 504^2} = \sqrt{9^2 \cdot 36^2 + 126^2}$$

... So the largest possible real part of the expression is \boxed{1134}.

---

Table 3: List of emergent attention heads found through circuits. `L` and `H` refers to the layer and head indices, respectively. Circuits are constructed using AIME'24 benchmark as input. For each post-training methods, we describe newly emergent attention heads. Visualization of total reasoning heads aggregation in single model architecture is in Figure 2.

| Post-Training | List of Emergent Attention Heads in Circuits | # of Heads |
|---|---|---|
| Qwen-2.5-Math-1.5B (Baseline) | `L0H7, L21H10, L2H6, L11H1, L14H10 ...` | 56 |
| DeepSeek-R1-Distill-Qwen-1.5B | `L5H0, L5H2, L5H4, L6H10, L7H7 ...` | 32 |
| SFT with OpenR1-Math-220k | `L0H8, L11H3, L3H3, L5H1, L7H3 ...` | 34 |
| GRPO with OpenR1-Math-220k | `L0H8, L5H1, L7H1, L18H11, L11H8 ...` | 19 |
| GRPO with GSM8K | `L0H8, L5H1, L7H2, L3H3, L21H2 ...` | 20 |

Table 4: Reasoning Head Ablation Inference for Qwen2.5-Math-1.5B and 7B. Every performance is measured with pass@1 score with temperature 0.6. Each ablation cases make the value of specific attention heads, around 5 number of heads from its circuit results, into zero for checking its importance for reasoning tasks. We color some scores into red which is the most degraded results except no ablation baseline, while the bold is the completely ruined performance. We also color performance increase with green when its heads are ablated. Overall tendency is reversed from Table 1, as base model heads are more effective than reasoning heads when ablated.

| Model | Method | AIME'24 | AIME'25 | GPQA | AMC |
|---|---|---|---|---|---|
| Qwen2.5 Math-1.5B | No Ablation | 13.3 | 4.73 | 9.74 | 38.5 |
| | Ablation with Reasoning Heads | 9.01 | 4.58 | 7.82 | 35.6 |
| | Ablation with Base Model Heads | 8.33 | 4.63 | 9.79 | 34.2 |
| | Ablation with TriviaQA Heads | 0.05 | **0.00** | 5.38 | 3.42 |
| Qwen2.5 Math-7B | No Ablation | 13.3 | 10.0 | 15.1 | 32.5 |
| | Ablation with Reasoning Heads | 6.67 | 10.0 | **20.2** | **43.3** |
| | Ablation with Base Model Heads | **23.3** | 3.33 | 15.6 | **43.3** |
| | Ablation with TriviaQA Heads | 20.0 | 10.0 | 16.1 | 37.3 |

Table 5: Head Intervention Inference for Qwen2.5-Math-1.5B with SFT and GRPO heads. Every performance is measured with pass@1 score with temperature 0.6. Each ablation cases make the value of specific attention heads, around 5 number of heads from its circuit results, into zero for checking its importance for reasoning tasks. Scale up cases increase the activation of specific attention heads into 1.3 higher, while scale down decrease it into half (0.5). We color some scores into red which is the most degraded results except no ablation baseline, while the bold is the completely ruined performance. We also color performance increase with green when its heads are ablated.

| Model | Method | AIME'24 | AMC | MATH |
|---|---|---|---|---|
| Qwen2.5-Math-1.5B | No Ablation | 13.3 | 38.5 | 56.0 |
| | Ablation with SFT Heads | **0.00** | 0.05 | 0.10 |
| | Scale Up with SFT Heads | **0.00** | 37.3 | 58.2 |
| | Scale Down with GRPO GSM8K Heads | 3.33 | **42.1** | **63.0** |
| | Scale Up with GRPO GSM8K Heads | 3.33 | 30.1 | **60.2** |

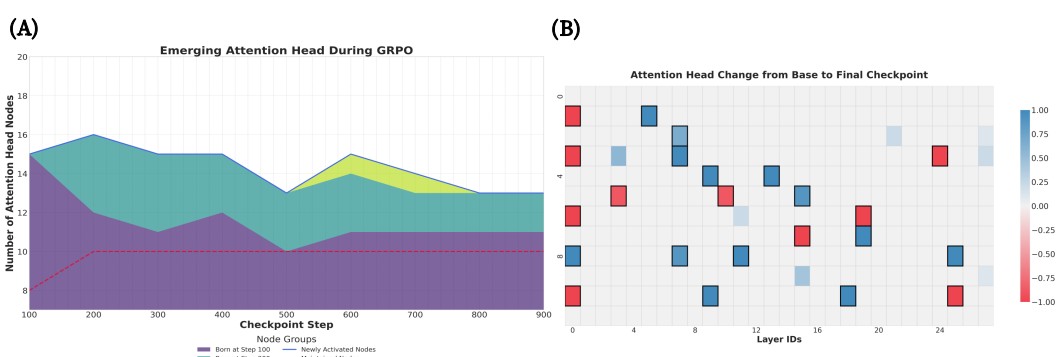

Figure 7: Analysis of Emergent Attention Head in Qwen2.5-Math-1.5B during GRPO with GSM8k (Cobbe et al., 2021) dataset, and circuit construction with AIME (AIME, 2025) benchmark. (A) denotes a cohort analysis of attention head activation over training checkpoints. The blue line tracks the absolute number of newly activated heads compared to the base model, while the red dashed line indicates the number of original heads that are maintained. The stacked areas represent cohorts of heads, color-coded by the checkpoint at which they first emerged, showing their persistence and evolution over time. (B) shows a heatmap detailing the changes in activation frequency. Red cells denote heads from the original base model, with fading intensity indicating their gradual deactivation. Blue cells represent newly emerged heads, with darker shades signifying higher activation frequency across checkpoints. Heads active in the final checkpoint are outlined with a black border.

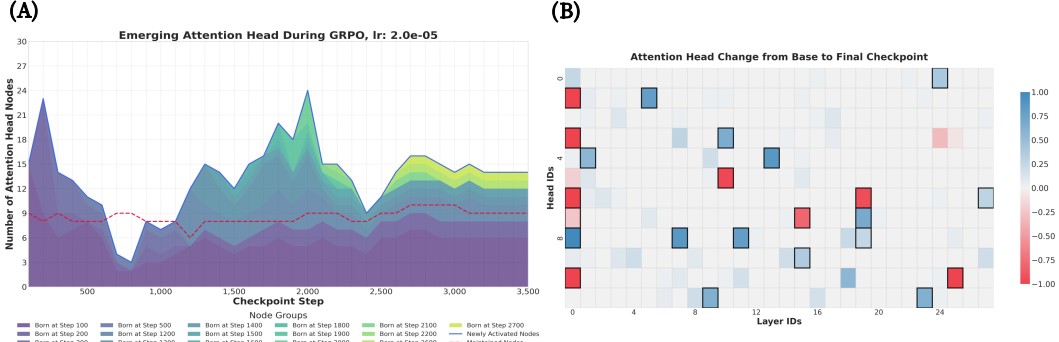

Figure 8: Analysis of Emergent Attention Head in Qwen2.5-Math-1.5B during GRPO with OpenR1-Math-220k (Hugging Face, 2025) dataset with learning rate 2e-05, and circuit construction with AIME (AIME, 2025) benchmark. (A) denotes a cohort analysis of attention head activation over training checkpoints. The blue line tracks the absolute number of newly activated heads compared to the base model, while the red dashed line indicates the number of original heads that are maintained. The stacked areas represent cohorts of heads, color-coded by the checkpoint at which they first emerged, showing their persistence and evolution over time. The fluctuation in newly activated heads shows a similar trend to the (B), accuracy reward curve. (C) shows a heatmap detailing the changes in activation frequency. Red cells denote heads from the original base model, with fading intensity indicating their gradual deactivation. Blue cells represent newly emerged heads, with darker shades signifying higher activation frequency across checkpoints. Heads active in the final checkpoint are outlined with a black border.

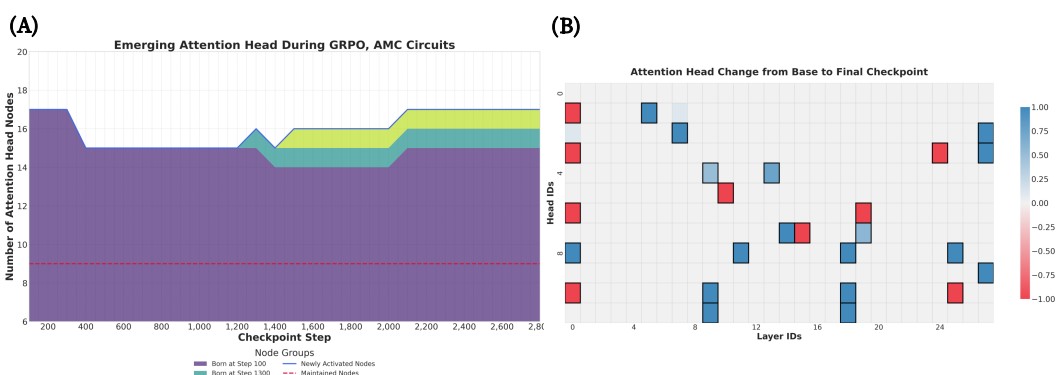

Figure 9: Analysis of Emergent Attention Head in Qwen2.5-Math-1.5B during GRPO with OpenR1-Math-220k (Hugging Face, 2025) dataset, and circuit construction with AMC (AI-MO, 2024) benchmark. (A) denotes a cohort analysis of attention head activation over training checkpoints. The blue line tracks the absolute number of newly activated heads compared to the base model, while the red dashed line indicates the number of original heads that are maintained. The stacked areas represent cohorts of heads, color-coded by the checkpoint at which they first emerged, showing their persistence and evolution over time. (B) shows a heatmap detailing the changes in activation frequency. Red cells denote heads from the original base model, with fading intensity indicating their gradual deactivation. Blue cells represent newly emerged heads, with darker shades signifying higher activation frequency across checkpoints. Heads active in the final checkpoint are outlined with a black border.

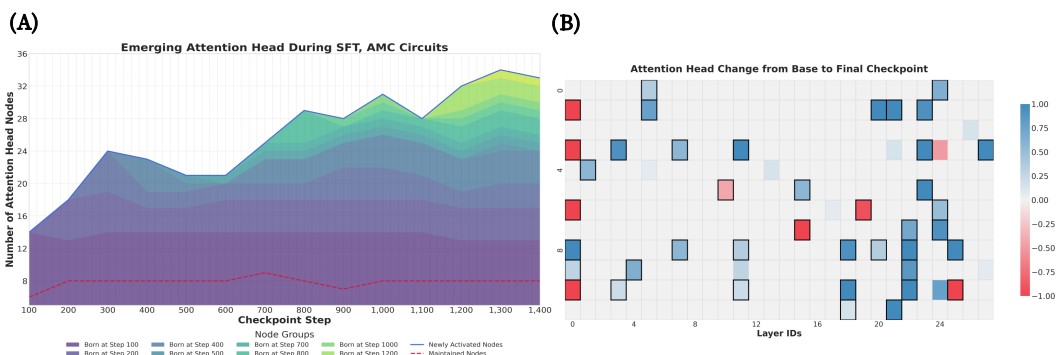

Figure 10: Analysis of Emergent Attention Head in Qwen2.5-Math-1.5B during SFT with OpenR1-Math-220k (Hugging Face, 2025) dataset, and circuit construction with AMC (AI-MO, 2024) benchmark. (A) denotes a cohort analysis of attention head activation over training checkpoints. The blue line tracks the absolute number of newly activated heads compared to the base model, while the red dashed line indicates the number of original heads that are maintained. The stacked areas represent cohorts of heads, color-coded by the checkpoint at which they first emerged, showing their persistence and evolution over time. The fluctuation in newly activated heads shows a similar trend to the (B), accuracy reward curve. (C) shows a heatmap detailing the changes in activation frequency. Red cells denote heads from the original base model, with fading intensity indicating their gradual deactivation. Blue cells represent newly emerged heads, with darker shades signifying higher activation frequency across checkpoints. Heads active in the final checkpoint are outlined with a black border.

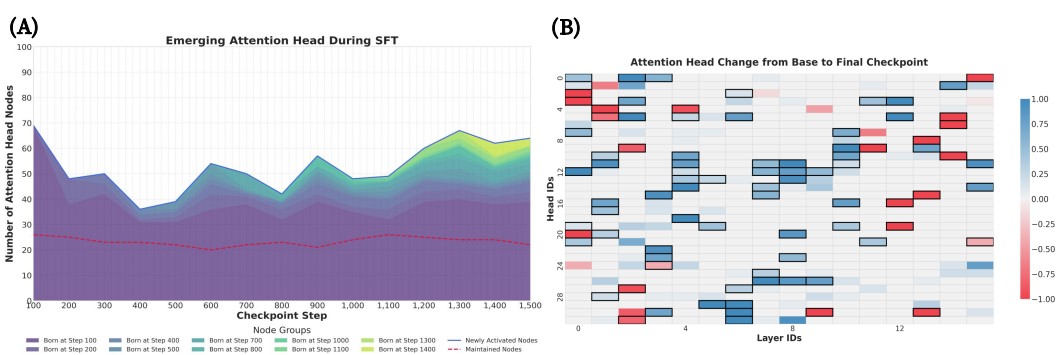

Figure 11: Analysis of Emergent Attention Head in Llama-3.2-1B-Instruct during SFT, trained with OpenR1-Math-220k (Hugging Face, 2025) and constructed circuit with AIME (AIME, 2025). (A) denotes a cohort analysis of attention head activation over training checkpoints. The blue line tracks the absolute number of newly activated heads compared to the base model, while the red dashed line indicates the number of original heads that are maintained. The stacked areas represent cohorts of heads, color-coded by the checkpoint at which they first emerged, showing their persistence and evolution over time. (B) shows a heatmap detailing the changes in activation frequency. Red cells denote heads from the original base model, with fading intensity indicating their gradual deactivation. Blue cells represent newly emerged heads, with darker shades signifying higher activation frequency across checkpoints. Heads active in the final checkpoint are outlined with a black border.

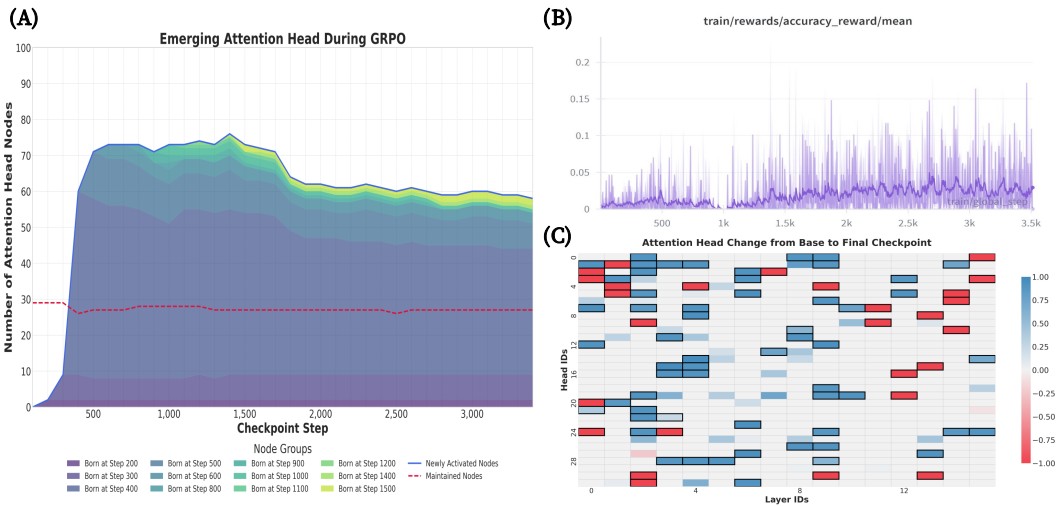

Figure 12: Analysis of Emergent Attention Head in Llama-3.2-1B-Instruct during GRPO, trained with OpenR1-Math-220k (Hugging Face, 2025) and constructed circuit with AIME (AIME, 2025). (A) denotes a cohort analysis of attention head activation across trained checkpoints. The blue line tracks the absolute number of newly activated heads compared to the base model, while the red dashed line indicates the number of original heads that are maintained. The stacked areas represent cohorts of heads, color-coded by the checkpoint at which they first emerged, showing their persistence and evolution over time. The fluctuation in newly activated heads shows a similar trend to the (B), accuracy reward curve. (C) shows a heatmap detailing the changes in activation frequency. Red cells denote heads from the original base model, with fading intensity indicating their gradual deactivation. Blue cells represent newly emerged heads, with darker shades signifying higher activation frequency across checkpoints. Heads active in the final checkpoint are outlined with a black border.

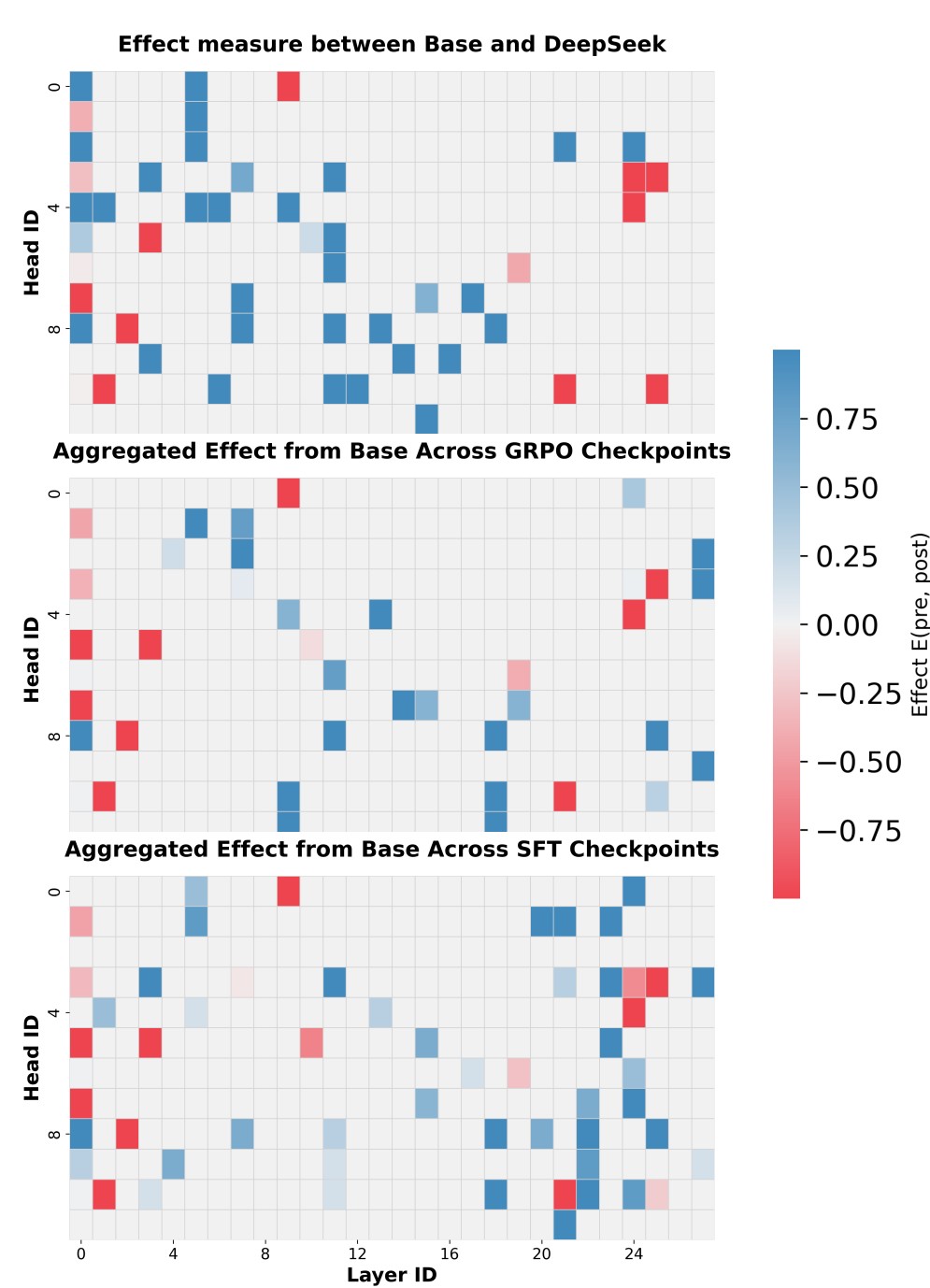

Figure 13: Head-level effect maps for Qwen2.5-Math-1.5B and its post-trained variants. From top to bottom: Effect between the base Qwen2.5-Math-1.5B model and the DeepSeek-distilled reasoning model; Effect aggregated across GRPO checkpoints (500-step intervals from 500 to 2500 steps) trained from the same base; Effect aggregated across SFT checkpoints (200-step intervals). Each cell corresponds to an attention head $(\ell, h)$, and the color encodes the symmetric effect measure $E(\ell, h) = \big(I_{\text{post}}(\ell, h) - I_{\text{pre}}(\ell, h)\big)/\big(I_{\text{post}}(\ell, h) + I_{\text{pre}}(\ell, h) + \varepsilon\big)$, where $I_{\text{pre}}$ and $I_{\text{post}}$ are the EAP-IG–based head importances defined in §A.4. Blue (red) indicates increased (decreased) attribution-based importance of the head relative to the base model. The high-magnitude heads in these maps qualitatively align with the high-frequency circuit heads in Figure 3 (B) and 4 (C), indicating that our frequency-based circuit analysis is consistent with the attribution-based importance view.

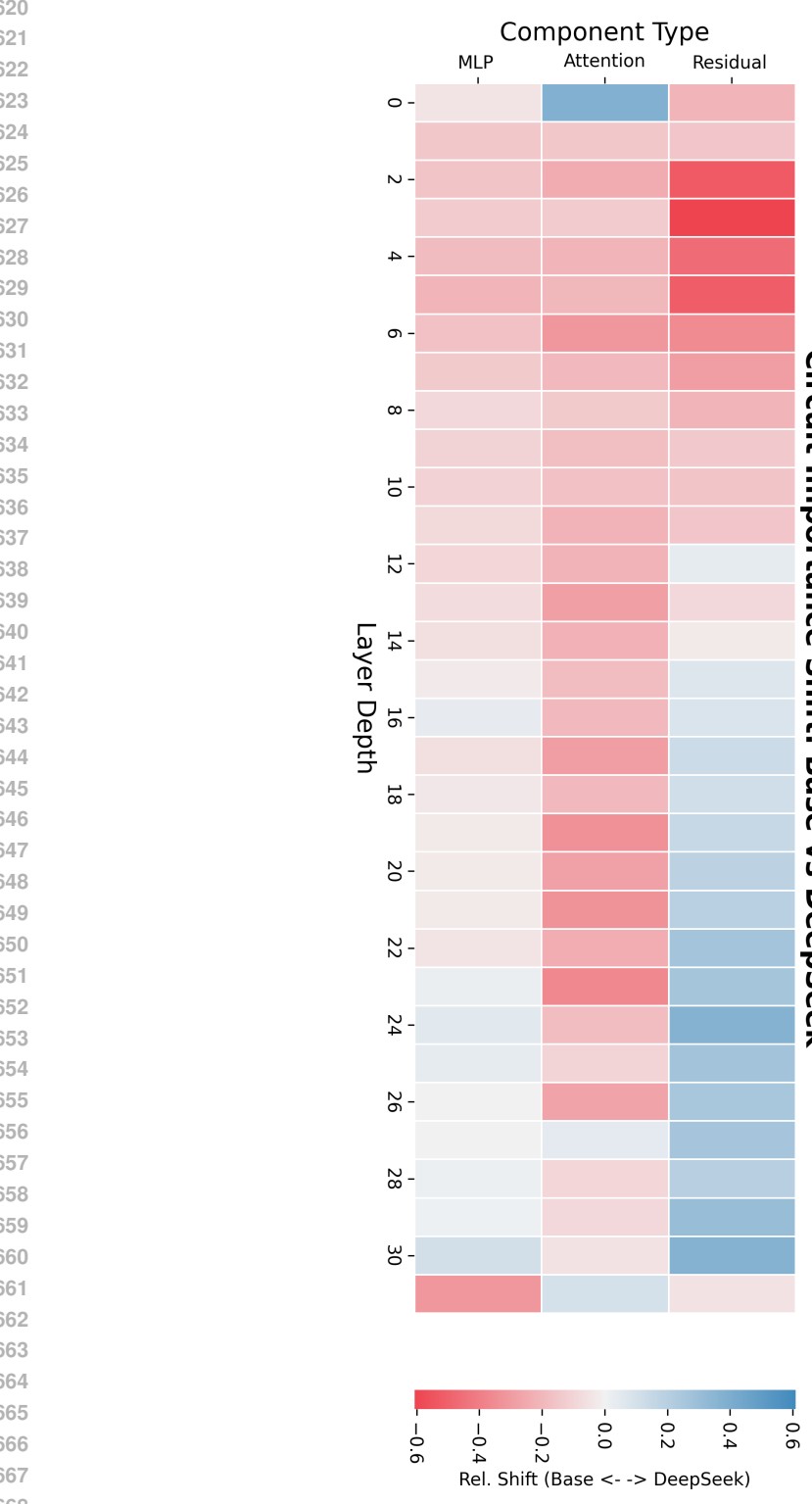

Figure 14: Component-level importance shift between Llama-3.1-8B (Base) and DeepSeek-R1-Distill-Llama-8B derived from Sparse Feature Circuits. Columns represent the aggregated attribution score for MLP, Attention, and Residual components across layers. The color encodes the symmetric effect measure. Blue (positive) indicates components where the DeepSeek model places higher causal weight (e.g., Layer 0 Attention and late-stage Residual streams), while Red (negative) indicates components more dominant in the Base model.

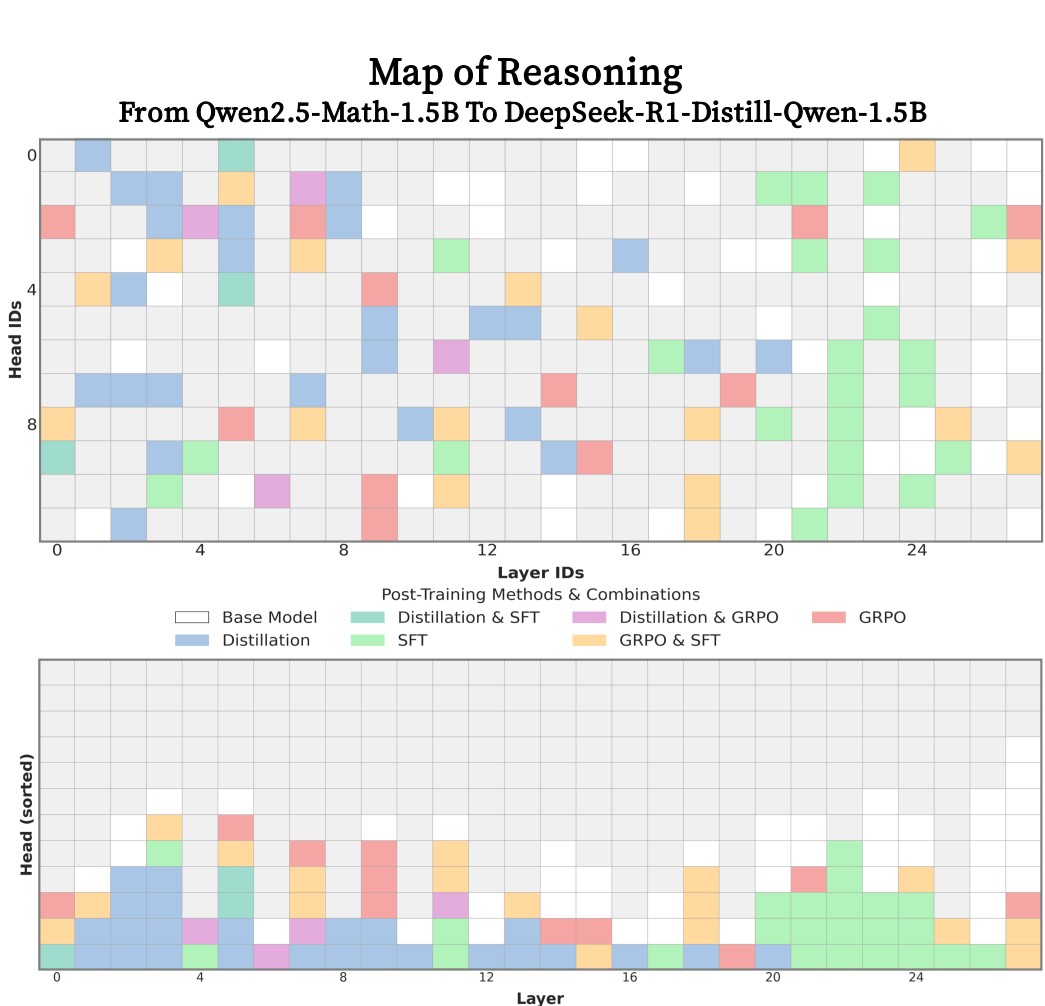

Figure 15: Map of Reasoning: Visualization of emergent reasoning heads in circuits based on Qwen2.5-Math-1.5B with various post-training, and DeepSeek-R1-Distill-Qwen-1.5B. (Top) A map of emergent attention heads for each post-training method, compared to the baseline model (white). (Bottom) A cumulative map of the reasoning heads, with columns sorted by the number of newly activated heads. Each GRPO and SFT category encompass both AIME and AMC benchmark based circuits, with checkpoints of both training using OpenR1-Math-220k and GSM8k dataset. DeepSeek Distillation activates enormous heads (blue), as SFT activates similarly large amount of heads, though SFT heads are mostly concentrated in mid-to-late layer (green). Some of attention heads from GRPO training are also common in the SFT and Distillation reasoning heads (yellow and purple), however, the number of GRPO heads are much smaller and distributed across layers (red).

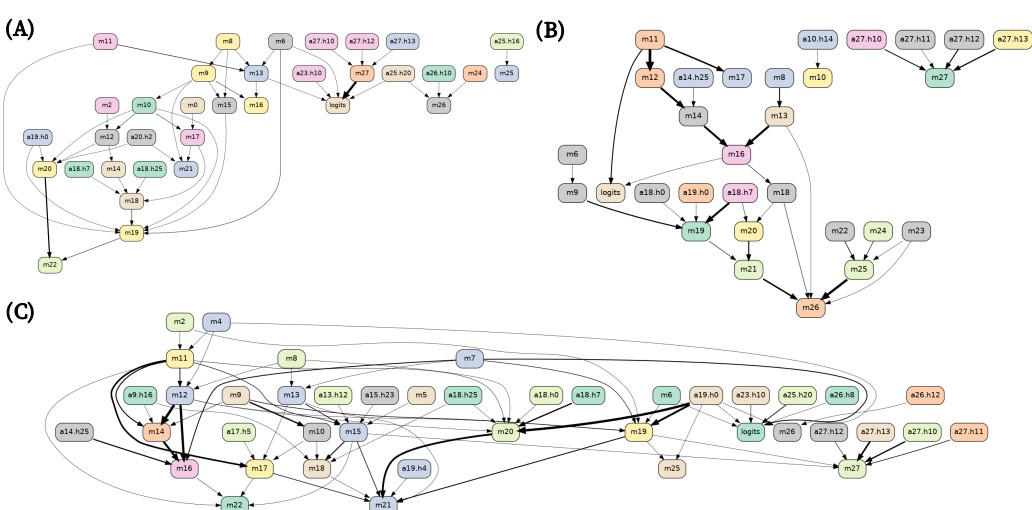

Figure 16: Actual Example of Circuits. Color of nodes are randomly mapped to differentiate each others. (A) denotes AIME circuit with baseline model, Qwen-2.5-Math-7B. (B) shows AIME circuit with DeepSeek-R1-Distill-Qwen-7B. (C) is the comparative example with same AIME dataset, which is constructed with DeekSeek-R1-Distill-Qwen-7B and its own sampled answer, without explicit <think>. (C) is more complex than other two circuits, which could be mixed with confusable attention heads. The trend of this enormous attention heads in (C) is also similar with the thinking off mode in Figure 18 (B), where the model compensate its performance gap through large emergent attention heads.

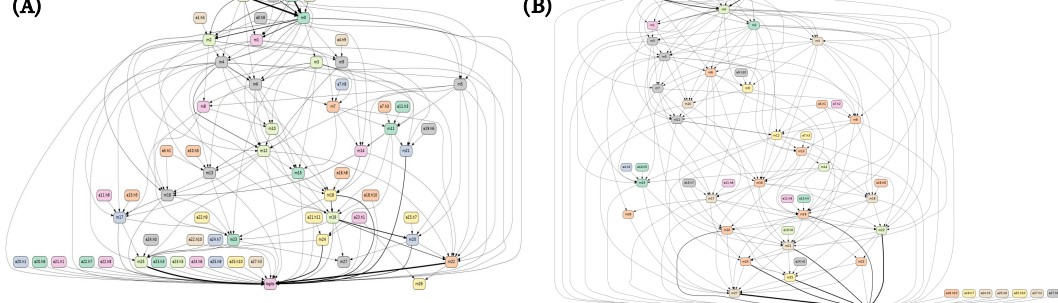

Figure 17: Actual Example of Circuits After Post-Training. Color of nodes are randomly mapped to differentiate each others. (A) denotes AIME circuit after SFT with baseline model, Qwen-2.5-Math-1.5B. (B) shows AIME circuit after GRPO with the same baseline model. (A) activates more attention heads while (B) has more complexly connected specific nodes which refer its internalized high-level mathematical reasoning.

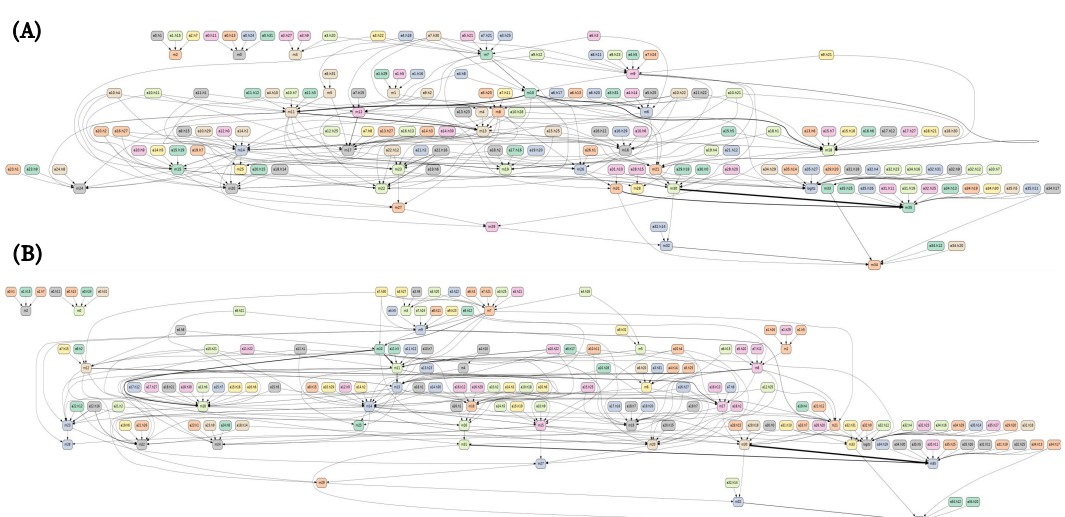

Figure 18: Actual Example of Circuits of Think On/Off. Color of nodes are randomly mapped to differentiate each others. (A) denotes AIME circuit of Thinking on mode in Qwen3-8B. (B) shows AIME circuit of Thinking off on the same baseline model. (B) activates more attention heads, in contrast, (A) has more complexly connected specific nodes which refer its internalized high-level mathematical reasoning, similar as GRPO circuit in Figure 17 (B).

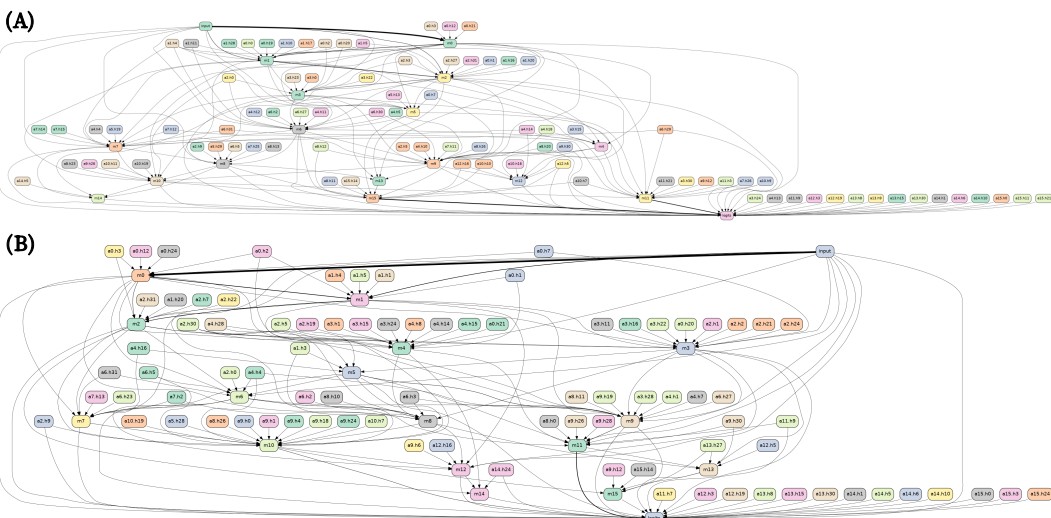

Figure 19: Actual Example of Circuits of Llama-3.2-1B-Instruct. Color of nodes are randomly mapped to differentiate each others. (A) denotes AIME circuit of Llama 3.2 after SFT with OpenR1-Math-220k dataset. (B) shows AIME circuit of Llama 3.2 after GRPO with OpenR1-Math-220k dataset.

