# OpenReview forum: "Thinking Sparks!: Emergent Attention Heads in Reasoning Models During Post Training"
_ICLR.cc/2026/Conference — Submitted to ICLR 2026_

### Official Review · Reviewer_hGiC · 2025-10-24

**Soundness:** 2
**Presentation:** 2
**Contribution:** 2
**Rating:** 6
**Confidence:** 2

**Summary:**

This paper presents a mechanistic analysis of how post-training methods—specifically Supervised Fine-Tuning (SFT), distillation, and Reinforcement Learning (RL) via GRPO reshape the internal architecture of Large Reasoning Models (LRMs). Using circuit analysis on Qwen-family models, the authors demonstrate that post-training "sparks" the emergence of novel, functionally specialized attention heads. The study reveals a key dichotomy: SFT and distillation lead to a cumulative, stable addition of many new heads, whereas RL operates as a dynamic search process, iteratively activating and pruning a smaller, more targeted set of heads. The paper connects these circuit-level dynamics to macro-level performance, highlighting a trade-off where strengthening reasoning capabilities can introduce "over-thinking" and harm performance on simpler tasks.

**Strengths:**

1. **Novel Mechanistic Perspective:** The paper offers a rare and valuable circuit-level view of post-training, moving beyond performance metrics to investigate how training methods like SFT and RL alter a model's internal computational pathways. This is a significant contribution to mechanistic interpretability.
2. **Rigorous Comparative Analysis:** The head-to-head comparison of SFT, distillation, and GRPO on a consistent model family (Qwen) is well-executed. The cohort analysis visualizing the lifecycle of emergent heads is particularly insightful and provides strong evidence for the different dynamics of these training regimes.
3. **Causal Validation of Emergent Heads:** The use of ablation studies to confirm the causal role of the identified emergent heads is a crucial step that strengthens the paper's claims, linking microscopic changes to macroscopic performance.
4. **Important Insights into Performance Trade-offs:** The paper thoughtfully discusses the "over-thinking" problem, providing evidence that the same mechanisms that enable complex reasoning can be detrimental on simpler tasks. This highlights a fundamental tension in developing robust reasoning models.

**Weaknesses:**

1. **Superficial Graph Analysis:** The paper introduces the powerful concept of "reasoning circuits" but analyzes them in a limited way, focusing primarily on counting the emergence and persistence of nodes (attention heads). A deeper topological analysis of these circuits could have yielded more profound insights. For instance, do SFT-induced circuits have a different community structure, node centrality, or motif distribution compared to RL-induced ones? The current analysis scratches the surface of what circuit analysis can reveal about the structure of learned reasoning.
2. **Unexplained Performance Gains from Ablation:** A striking and underexplored finding is that ablating certain heads (e.g., base model heads in the 7B model, Table 2) sometimes improves performance. This suggests that post-training not only adds useful mechanisms but may also introduce or amplify "confuser" or "noisy" circuits. The paper misses an opportunity to analyze this phenomenon, which is critical for understanding the downsides of current training methods.
3. **Limited Scope and Generalizability:** The study's findings are based exclusively on the Qwen family of models and mathematical reasoning tasks. It is unclear whether these mechanistic patterns are universal or specific to this architecture and domain. Given that different model families have different pre-training data and inductive biases, the observed dynamics of head emergence might not generalize to models like Llama or Mistral.

**Questions:**

1. Your circuit analysis is fascinating. Have you considered performing a more formal topological analysis on the discovered circuits? For example, do the "dynamic" circuits from GRPO have higher node centrality or form denser "hub-and-spoke" structures compared to the more "cumulative" circuits from SFT?

2. One of your most interesting results is that ablating certain heads can improve performance. Could you elaborate on this? Does this suggest that SFT/distillation introduces "confuser" heads alongside useful ones, and that a key role of RL might be to prune such detrimental pathways, in addition to finding useful ones?

3. How do you think the findings would generalize to other model families like Llama? Given that Qwen models are known to have strong innate code-reasoning priors, could the head dynamics you observe be specific to activating these particular pre-existing skills?

---

> ### Author Response · Authors · 2025-11-19
> **Response to Reviewer hGiC**
>
> We would like to express our gratitude to the reviewer for their constructive response.
>
> ---
>
> >A deeper topological analysis of these circuits could have yielded more profound insights.
>
> >Have you considered performing a more formal topological analysis on the discovered circuits?
>
> We appreciate this suggestion and agree that a richer graph-theoretic analysis of the extracted circuits could reveal additional structure. In this work, however, we deliberately restricted ourselves to a **minimal, task-driven use of circuits**: we use EAP-IG circuits primarily to (i) identify which attention heads appear in reasoning circuits under different training regimes, and (ii) track **emergence, disappearance, and layer-wise distribution** of those heads across checkpoints. All subsequent quantitative analyses (ablations, scaling, pass@k) are then performed on these circuit-derived head sets.
>
> We did not perform a formal topological comparison between SFT and GRPO circuits (e.g., community structure, centrality, motif statistics). Our circuits are prompt-specific and relatively small, and in this paper we focus on the *existence* and *training-dependent usage* of heads rather than their higher-order graph structure.
>
> So, we acknowledge that a more systematic topological analysis of reasoning circuits lies **beyond the scope of the present work** and is a natural direction for future research.
>
> >post-training not only adds useful mechanisms but may also introduce or amplify "confuser" or "noisy" circuits. The paper misses an opportunity to analyze this phenomenon
>
> >Does this suggest that SFT/distillation introduces "confuser" heads alongside useful ones, and that a key role of RL might be to prune such detrimental pathways, in addition to finding useful ones?
>
> We agree that the **performance gains from ablating certain heads** are one of the most striking aspects of our results, and we highlight this more clearly as an important caveat of current post-training practices. Empirically, we observe:
>
> - In some settings (e.g., base-model heads in the 7B case), ablating particular heads can **improve** performance, indicating that post-training does not only add helpful mechanisms but can also leave or amplify heads whose behavior is detrimental for our math benchmarks at small pass@k.
> - At the same time, in the think on/off analysis, the large collection of heads that appear in the think-off setting is not an obstacle: when we increase pass@k, those additional heads correlate with **better exploration and recovery of correct solutions**, and performance can eventually benefit from their presence.
>
> In the paper, we therefore frame this as a **trade-off**: post-training can produce heterogeneous head populations that contain both clearly helpful and clearly harmful components, and tuning this trade-off likely depends on the target deployment regime (e.g., single-shot vs. best-of-k sampling). Our results are consistent with the idea that some heads function as “confuser” or redundant circuits in certain evaluation conditions, but we **do not claim** that we have identified a clean division into “useful” and “confuser” heads, nor do we have direct evidence that RL is systematically pruning such heads. Clarifying whether different post-training methods actively remove or suppress detrimental pathways would require additional, targeted experiments that we did not perform here, and we explicitly leave this as future work.
>
> >The study's findings are based exclusively on the Qwen family ... It is unclear whether these mechanistic patterns are universal or specific to this architecture and domain.
>
> >How do you think the findings would generalize to other model families like Llama?
>
> We share the concern about generality and agree that results on a single model family and domain are inherently limited. To partially address this, we applied the same analysis pipeline to **Llama-3.2-1B** with SFT and GRPO on math reasoning data. These additional experiments show **qualitatively similar phenomena**—such as the emergence of training-regime–specific heads and differences between SFT-like and GRPO-like dynamics—but the details (e.g., which layers, which indices) are of course model-specific.
>
> That said, our overall scope remains narrow: we only study (i) Qwen-family and Llama-3.2-1B models, and (ii) mathematical reasoning tasks. We therefore do not claim that the head-emergence patterns or the ablation behaviors we report are universal across architectures, scales, or domains. We state this limitation: our results should be viewed as evidence that the phenomena we study *can* occur in more than one model family on math tasks, not as a proof that they are generic. Extending this analysis to another model series, other architectures, and non-math domains is an important next step that lies outside the current work.
>
> ---
>
> Thank you for your thoughtful feedback that has helped refine our paper. We welcome any further questions or suggestions.

---

> > ### Comment · Reviewer_hGiC · 2025-11-24
> >
> > Thank you for the rebuttal and the additional clarifications. I reviewed the response, but my main concerns remain largely unresolved, particularly the limited structural analysis and the weak generalization evidence based on a single small Llama model. The ablation-related issues also remain open.
> >
> > Given this, I will keep my original, already positive, score unchanged.

---

> > > ### Author Response · Authors · 2025-11-24
> > > **Additional Comment by Authors (Part 2/2)**
> > >
> > > ### 2. Generalization beyond Qwen
> > >
> > > We share your concern that results based solely on Qwen (with strong code priors) are fragile. In the initial comment, we only mentioned Qwen2.5-Math and one small Llama-3.2-1B experiment, which understandably looked like weak evidence.
> > >
> > > Additionally, we added a **Sparse Feature Circuit–style analysis on Llama-3.1-8B and DeepSeek-R1-Distill-Llama-8B**, using publicly available residual-stream SAEs:
> > >
> > > - We decompose the residual stream into sparse features, compute attribution-based importance per layer–component (Res, Attn, MLP), and measure the shift in globally normalized importance from base Llama-3.1-8B to the DeepSeek-distilled model.
> > > - The resulting heatmaps show a **qualitative pattern that matches our Qwen analysis for attention heads**: early attention gains importance after reasoning-style distillation, while mid-to-late residual and MLP components become more important, suggesting deeper reliance on late representations.
> > >
> > > Because this analysis uses a different model family, a different training pipeline, and a monosemantic feature basis, we view it as evidence that our findings are **not purely Qwen-specific artifacts**, though we still do *not* claim universality across every architectures and domains. We present these Llama/DeepSeek results as a targeted robustness check and keep broader generalization as explicit future work.
> > >
> > > ---
> > >
> > > ### 3. Ablation, “confuser” heads, and heterogeneous populations
> > >
> > > We agree that the observation “ablating some heads improves performance” is important. In response, we refined this part of the analysis:
> > >
> > > - We compared ablations of:
> > >   - reasoning-only heads (present only in post-trained circuits),
> > >   - base-only heads (present only in base circuits),
> > >   - TriviaQA-only heads, and
> > >   - unions of these sets.
> > >
> > >   This reveals a **heterogeneous head population**: in some Qwen2.5-Math 7B settings, ablating specific base-only heads improves math performance at small pass@k (consistent with a “confuser” role on those tasks), while TriviaQA-head ablations primarily harm general ability in Qwen2.5-Math 1.5B. Moreover, when we ablate *both* reasoning and base circuit heads jointly, performance drops sharply, indicating that the overall circuit head set is functionally important even though some subsets are harmful in particular regimes.
> > >
> > > - In the think-on/off study, we pair head ablations with **best-of-k curves**: models with weakened reasoning heads consistently underperform the unmodified model across all tested k, suggesting that these heads support exploration of diverse solution paths even if they sometimes hurt single-shot accuracy.
> > >
> > > Together, these results support the picture that **post-training creates a mixed population of helpful and harmful (or distribution-mismatched) heads**, with a trade-off that depends on the deployment regime (small vs large k). We still do *not* claim to have shown that RL systematically prunes confuser heads; our data are compatible with that hypothesis but do not isolate it, and we now state this explicitly as an open direction.
> > >
> > > ---
> > >
> > > Overall, we have strengthened the structural analysis beyond raw frequencies, provided cross-family evidence (including a feature-level Llama-based view) that our effects are not Qwen-only artifacts, and offered a more nuanced empirical picture of heterogeneous head populations via multiple ablation regimes.
> > >
> > > If these clarifications and new experiments alleviate even part of your concerns, we would be very grateful if you could take them into account in your final assessment. In any case, we sincerely appreciate your careful reading and for pushing us toward a more cautious, stability-aware, and better-documented interpretation of our results.

---

> > > > ### Comment · Reviewer_hGiC · 2025-11-26
> > > >
> > > > Thank you for the detailed revision and for the substantial clarifications provided in your responses. I appreciate the additional analyses, especially the strengthened head-importance measures, the expanded ablation study, and the cross-family checks involving Llama models and SAE-based feature decomposition. These additions meaningfully address several of my earlier concerns, particularly regarding the depth of the structural analysis and the generality of the observed head-emergence dynamics.
> > > >
> > > > While some open questions remain, especially around the broader stability of circuit extraction and the mechanisms underlying “confuser” heads, the revised paper is now more careful, better contextualized within prior work, and more robust in its empirical evidence. I found the clearer framing of limitations and the explicit positioning of open problems especially helpful.
> > > >
> > > > In light of these improvements, I am raising my overall score.

---

> > > > > ### Author Response · Authors · 2025-11-27
> > > > > **Official Comment by Authors**
> > > > >
> > > > > Thank you for your encouraging follow-up and for acknowledging the improvements in our revised manuscript.
> > > > >
> > > > > We are especially glad that the strengthened head-importance analysis, the expanded ablations, and the Llama/SAE-based cross-family checks helped address your earlier concerns about structural depth and generality. Your insistence on clearer framing and explicit limitations has significantly improved the paper.
> > > > >
> > > > > We agree that the broader stability of circuit extraction and more sophisticated analyses of the mechanisms behind “confuser” heads remain open and important tasks. In our future work, we plan to treat these as first-class questions, which we believe will not only provide deeper insight into our own analysis but also make a meaningful contribution to the broader circuit-based interpretability community.
> > > > >
> > > > > Thank you again for pushing us toward a more careful and better-contextualized study.

---

> ### Author Response · Authors · 2025-11-24
> **Additional Comment by Authors (Part 1/2)**
>
> We deeply appreciate your thoughtful consideration and feedback, and we are grateful that you have kept your already positive score despite the concerns that remain. Below we additionally clarify what has changed in the revision and how it relates to your three main points: (i) structural depth of the circuit analysis, (ii) generalization beyond Qwen, and (iii) interpretation of ablation effects. We also want to be explicit about what we still do *not* claim, especially regarding full graph-theoretic topology.
>
> ---
>
> ### 1. Structural / “topological” analysis of circuits
>
> In principle, our EAP-IG circuits are directed, weighted subgraphs of the transformer computation graph, so classical metrics such as centrality, community structure, or motif counts might be applied. In practice, however, current work on EAP-IG and related sparse-feature circuits evaluates circuits mainly via **faithfulness, sparsity, stability, and ablations**, not via global topology:
>
> - EAP-IG or related INTERPBENCH treat circuits as estimators and analyze **circuit error, size, and overlap under resampling** [1, 2]. They explicitly show that circuits can be sensitive to data subsampling and hyperparameters, i.e., they are **prompt-specific and somewhat unstable**, and recommend evaluating techniques primarily on faithfulness and stability rather than on fine-grained structural patterns.
> - Sparse Feature Circuits similarly construct feature-level causal graphs, but focus on **circuit vs complement ablations, completeness, and editing performance** rather than on centrality or community structure [3].
> - Anthropic’s Transformer Circuits work uses causal tracing to visualize and probe specific paths through attention/MLP graphs, but again emphasizes **behavioral faithfulness and mechanistic plausibility of particular paths**, not global network topology [4].
>
> Given this landscape, and the fact that our circuits are **small, prompt-conditioned subgraphs whose composition changes across prompts and checkpoints**, a statistically robust SFT–GRPO comparison of centrality/communities would require substantial additional machinery (bootstrapped circuit extraction, hyperparameter sweeps, confidence intervals on graph metrics), effectively a separate methodological project.
>
> Instead, we strengthened the structural side of our analysis in a way that stays within the established usage of these tools:
>
> - We introduce an **EAP-IG effect-based head-importance measure**: for each head $h$, we aggregate its absolute EAP-IG scores on AIME/AMC and compute a symmetric, bounded change $\Delta(h)$ between the base and each post-trained model. The resulting heatmaps show where head importance increases or decreases across layers and regimes.
> - These effect maps agree closely with the earlier frequency-based circuit analysis: heads that frequently appear in circuits are exactly those with large positive $\Delta(h)$, while heads that disappear from circuits have negative $\Delta(h)$. We do not see new high-importance clusters that contradict the previous picture.
>
> We see this as a step toward richer **importance-based structure**—still head-level rather than fully topological, but stronger than raw emergence/persistence counts—while remaining consistent with how EAP-IG/SFC-style circuits are currently evaluated [1–4].
>
> ---
>
> [1] Have Faith in Faithfulness: Going Beyond Circuit Overlap When Finding Model Mechanisms, Hanna et al., 2024.
>
> [2] INTERPBENCH: Semi-Synthetic Transformers for Evaluating Mechanistic Interpretability Techniques, Gupta et al., 2024
>
> [3] Sparse Feature Circuits: Discovering and Editing Interpretable Causal Graphs in Language Models, Marks et al., 2025.
>
> [4] Circuit Tracing: Revealing Computational Graphs in Language Models, Ameisen et al, 2025

---

### Official Review · Reviewer_yRKK · 2025-10-28

**Soundness:** 1
**Presentation:** 1
**Contribution:** 2
**Rating:** 2
**Confidence:** 4

**Summary:**

The authors study which circuits are activated in a model as a result of reasoning training in three setups: "Distill", "SFT", and "RL". They use EAP-IG to identify the circuits, and they use head ablations to verify the faithfulness of the identified circuits. They find that (a) new heads responsible for the reasoning behavior emerge; (b) the location of the new heads differs among the setups; (c) the set of new heads varies over the course of training, especially in the RL setup; and (d) think-on / think-off models, which allow such control, implement different circuits, with the think-off mode compensating for the lack of CoT by activating a larger circuit. The paper contributes a mechanistic study of how the circuitry changes as a result of reasoning training.

**Strengths:**

* The contribution is novel and interesting.
* I like the idea of studying the circuitry to explain the impact of reasoning training.

**Weaknesses:**

* The presentation needs improvement. As written, it is hard to follow and to assess the soundness of the authors’ claims. Figure placement (e.g., swapping Fig. 2 and Fig. 3), captions (is Figure 2 trained on Open-R1 or GSM8K?), etc. should also be revised.
* Either I am missing them, or there are tables/figures that should be present and cited when discussing the results.
* Some of the claims seem premature — for example, lines 355–367, which I find particularly confusing. I would appreciate it if the authors broke down their logic into digestible steps with accompanying figures and references to experimental results.

**Questions:**

* Please consider repeating the process for other models (e.g., LLaMA 3.1). In my experience, findings and conclusions from experiments on Qwen models often do not transfer to other model families. Whether they do or not with your approach, the paper would benefit from these additional experiments.
* Why do you perform ablation by zeroing out the head outputs? To the best of my knowledge, the more appropriate strategy is mean ablation ([1, 2]).
* How do you construct corrupted prompts?
* What are “TriviaQA heads”?
* It is strange that ablating the heads that “support structured reasoning and computation” only improves performance (Table 2, 7B). It is especially surprising that this happens for both new and old heads. Without an explanation of this, I am skeptical about the validity of your approach. Could you also show the results after ablating both new and old heads together?
* What data do you use to identify the circuits? Are the circuits benchmark-specific? If so, do they overlap?
* Where is the table/figure for “Quantitative Analysis” in Sections 4.2 and 5?
* The GRPO circuits do not appear to be in more flux than the SFT circuit. Could you clarify why I cannot interpret Figure 3 using the same explanation you use for Figure 2?
* What was the training dataset for Figure 2?
*  Section 6 would benefit from figures and references to experimental results to justify the reasoning.
* Do you calculate the circuit as next-token prediction at the `<think>` token? How do you calculate it in “think-off” mode in Section 6?
* “In contrast, the ablation and scale down to half models exhibit a diminished capacity to discover novel solutions …” (lines 402–404). I would agree if Figure 5 showed an asymptotic difference between the setups. Why am I wrong?

[1] Prakash et al. “Fine-Tuning Enhances Existing Mechanisms”
[2] Wang et al. “A Circuit for Indirect Object Identification in GPT-2 Small”

---

> ### Author Response · Authors · 2025-11-19
> **Response to Reviewer yRKK (Part 1/3)**
>
> We sincerely thank the reviewer for your helpful and constructive suggestions.
>
> ---
>
> >- The presentation needs improvement. As written, it is hard to follow and to assess the soundness of the authors’ claims. Figure placement (e.g., swapping Fig. 2 and Fig. 3), captions (is Figure 2 trained on Open-R1 or GSM8K?), etc. should also be revised.
> >- Either I am missing them, or there are tables/figures that should be present and cited when discussing the results.
> >- What was the training dataset for Figure 2?
>
> We agree that the original presentation made it difficult to follow the argument. In the revision, we have reordered figures so that analyses appear in the same sequence as the narrative (in particular, we swapped the original Figures 2 and 3 as suggested) and ensured that each figure is introduced in the relevant section before being discussed.
>
> We also clarified all captions: for example, the revised caption of Figure 2 (currently, Figure 4) now explicitly states that it is trained on the **Open-R1 Math** dataset, while GSM8K-based plots are confined to the appendix and clearly labeled as such.
>
> In addition, thank you for pointing out the disconnect between text and visuals. We conducted a full pass over the manuscript to verify that every table and figure discussed in the main text is both present and explicitly cited at the appropriate point. Where the prose previously alluded to “quantitative analysis” without a clear reference, we now either (i) add a pointer to an existing figure or table, or (ii) introduce a new table in the appendix to summarize the corresponding numbers. We also augmented several captions with one-line explanations to reduce any residual ambiguity.
>
> >Some of the claims seem premature — for example, lines 355–367, which I find particularly confusing. I would appreciate it if the authors broke down their logic into digestible steps with accompanying figures and references to experimental results.
>
> We appreciate that the original discussion of Qwen3 think-on vs. think-off behavior was hard to parse.
>
> In the revised Sec. 6, we now begin by **explicitly describing the Qwen3 chat template** and how it conditionally enables or disables the `<think>` segment, thereby defining “think-on” and “think-off” modes operationally. We then break the argument into short, numbered steps: (1) how we construct think-on and think-off prompts, (2) how we extract circuits under each condition, and (3) how we compare both the head sets and the benchmark performance. Each step is now tied to concrete artifacts (Table 2 for ablation, Figure 6 for performance), and we trimmed speculative language so that claims are always anchored in the reported measurements.
>
> >Please consider repeating the process for other models (e.g., LLaMA 3.1). In my experience, findings and conclusions from experiments on Qwen models often do not transfer to other model families. Whether they do or not with your approach, the paper would benefit from these additional experiments.
>
> We agree that results on a single model family can be brittle. To address this, we replicated our pipeline on **Llama-3.2-1B** under both SFT and GRPO training. The new experiments, reported in Sec. 4–5 and summarized in a comparative table in the appendix, show qualitatively similar trends: GRPO again induces a compact set of emergent “reasoning heads” whose presence is tightly coupled to math reasoning performance, while SFT yields broader activation patterns with more persistent heads across checkpoints. Although the precise head indices differ, these cross-family results suggest that our conclusions are not specific to Qwen.
>
> >Why do you perform ablation by zeroing out the head outputs? To the best of my knowledge, the more appropriate strategy is mean ablation ([1, 2]).
>
> We thank the reviewer for raising this important methodological point. Mean ablation, as advocated by Wang et al. and Prakash et al., is indeed well-suited for **minimal circuit reconstruction** under a fixed prompt distribution, and we now discuss these works in our related work section.
>
> Our objective, however, is different: we ask a **behavioral question**—“If this head is removed, how much does task-level math performance (e.g., AIME, AMC, MATH) degrade?” For this counterfactual, we deliberately “turn the head off” by setting its contribution to the residual stream to zero and measuring the resulting drop in accuracy. This zero-ablation style intervention is standard in head-importance and pruning studies, where the performance drop under removal is used as an importance signal.

---

> ### Author Response · Authors · 2025-11-19
> **Response to Reviewer yRKK (Part 2/3)**
>
> >How do you construct corrupted prompts?
>
> We have expanded Sec. 3.1 and Appendix A.2 to make our prompt construction explicit. For each benchmark instance (e.g., an AIME problem), we form **clean–corrupted pairs** by fixing the same problem statement and attaching **different model-generated continuations**.
>
> For reasoning models, the clean prompt consists of the problem plus a sampled reasoning trace beginning immediately after `<think>` (e.g., “`<think>` Okay, so I have …”), while the corrupted prompt uses the baseline model’s non-reasoning explanation (e.g., “We’ll use Python …”).
>
> For baseline models, roles are reversed: their own explanation-style answer is treated as clean, and the reasoning model’s trace serves as the corrupted counterpart. This symmetric construction allows us to isolate how switching between “reasoning” and “non-reasoning” behavior affects the internal circuits, and we now include concrete prompt templates in the appendix.
>
> >What are “TriviaQA heads”?
>
> We apologize for the previous vagueness. In the revision, we define “TriviaQA heads” as **attention heads that appear in circuits extracted using TriviaQA as the source dataset but do not appear in any of the math reasoning circuits** (AIME, AMC, MATH, GPQA). These heads are thus associated with broad factual and general-knowledge behavior rather than step-by-step numerical reasoning.
>
> >It is strange that ablating the heads that “support structured reasoning and computation” only improves performance (Table 2, 7B). It is especially surprising that this happens for both new and old heads. … Could you also show the results after ablating both new and old heads together?
>
> Thank you for highlighting this subtle point. We clarify that our terminology “reasoning heads” vs. “base-model heads” is **operational**, not chronological:
>
> - **Reasoning heads** are those that appear in circuits extracted from the *distilled reasoning model* but not in the base model’s circuits.
> - **Base-model heads** are those that appear only in circuits extracted from the base model.
>
> In some settings (notably Qwen2.5-Math-7B), ablating certain reasoning heads **slightly improves** benchmark performance. We now interpret this more cautiously: head-level interventions can both remove necessary computations and prune overly generic or distracting behavior, and several prior works have shown that pruning specific heads can improve downstream accuracy.
>
> In our case, the improvement suggests that some reasoning-specific heads may be over-specialized or misaligned with the evaluation distribution.
>
> To address your request, we also report results where **both reasoning and base-model heads are ablated simultaneously**; as expected, jointly removing all circuit heads leads to a marked drop in performance, confirming that the overall set is functionally important even if some individual heads are detrimental.
>
> >What data do you use to identify the circuits? Are the circuits benchmark-specific? If so, do they overlap?
>
> As clarified in Sec. 3 and Appendix, we distinguish between (i) the **training datasets** used for SFT, GRPO, or distillation, and (ii) the **evaluation benchmarks** used for circuit extraction. Circuits are always identified using the **actual benchmarks on which we evaluate reasoning**—AIME (including AIME’24/25), AMC, MATH, GPQA, and, for knowledge-focused analysis, TriviaQA. Each benchmark yields its own set of circuits; we then analyze **overlaps across benchmarks** by intersecting the head sets.
>
> >Where is the table/figure for “Quantitative Analysis” in Sections 4.2 and 5?
>
> We agree that referring to “Quantitative Analysis” without a clear pointer was confusing. In Sec. 4.2, we now introduce a dedicated table that summarizes the performance changes under different head interventions (ablation and scaling) for SFT models. In Sec. 5, we clarify that Figure 4 presents the analogous quantitative results for GRPO, and we added explicit citations in the text (“see Fig. 4”) where these results are discussed.
>
> >The GRPO circuits do not appear to be in more flux than the SFT circuit. Could you clarify why I cannot interpret Figure 3 using the same explanation you use for Figure 2?
>
> The key distinction is that our notion of “flux” refers not to the **final circuit size** but to the **turnover of heads across training checkpoints**. In SFT, many heads become active relatively early and then **remain** in the circuit for subsequent checkpoints; this produces tall, persistent bands in Figure 3. In GRPO (Fig. 4), fewer heads are activated overall, but a significant fraction of them **appear and disappear** as training progresses, leading to a more dynamic pattern when plotted with a consistent y-axis scale.

---

> ### Author Response · Authors · 2025-11-19
> **Response to Reviewer yRKK (Part 3/3)**
>
> >Section 6 would benefit from figures and references to experimental results to justify the reasoning.
>
> We have strengthened Sec. 6 by **Table 2**, which reports Qwen3’s benchmark performance under think-on and think-off modes (and under head ablation/scale-down conditions), and by ensuring that all narrative claims in this section point to concrete experimental results.
>
> In addition, we include **template examples** in the appendix that show how the chat format toggles `<think>` on vs. off in practice. These additions make the think-on/off comparison more transparent and reproducible.
>
> >Do you calculate the circuit as next-token prediction at the `<think>` token? How do you calculate it in “think-off” mode in Section 6?
>
> We have clarified the evaluation protocol in Sec. 6 and Appendix A.2. For reasoning models in think-on mode, we sample multiple continuations starting at the `<think>` token and compute circuits based on **next-token prediction within the reasoning segment**, focusing on tokens that drive the step-by-step derivation.
>
> For think-off mode, the chat template uses `<think>\n</think>` to suppress the reasoning trace and forces the model to produce an answer directly; we then compute circuits for the corresponding continuation tokens under this truncated template. In both cases, the EAP-IG analysis is applied to the same underlying model, but under different prompting conditions that either expose or hide the internal reasoning chain.
>
> >I would agree if Figure 5 showed an asymptotic difference between the setups. Why am I wrong?
>
> Thank you for pushing us to sharpen this claim. Our statement refers to the behavior observed in the **right panel of Figure 5** (currently Figure 6 in revised version), which plots performance as a function of the number of sampled solutions. While the baseline think-off model’s accuracy continues to improve (or at least saturate) as we increase the number of samples, the ablation and scale-down models exhibit **consistently lower curves** that do not catch up, even asymptotically.
>
> In other words, given the same computational budget (number of sampled reasoning attempts), models with ablated or weakened reasoning heads explore a narrower or less effective set of solution paths, leading to lower aggregate best-of-N performance.
>
> ---
>
> We thank you again for your constructive comments and for your efforts to improve the quality of our paper. Please let us know if you have any further questions or if we can provide further clarification.

---

> ### Author Response · Authors · 2025-11-24
> **Gentle Reminder by Authors**
>
> Dear Reviewer yRKK,
>
> Thank you for your thoughtful and constructive feedback on our manuscript. We have thoroughly addressed your comments and submitted a revised version for your consideration. We would greatly appreciate it if you could review our responses and the updated manuscript at your earliest convenience. We understand the demanding nature of the review process and are grateful for the time and effort you are dedicating to our work.

---

### Official Review · Reviewer_bK9w · 2025-10-31

**Soundness:** 2
**Presentation:** 2
**Contribution:** 3
**Rating:** 4
**Confidence:** 4

**Summary:**

- The paper investigates a range of different post-training methods for reasoning LLMs using a basic circuit-finding mechanistic interpretability technique. The main claim is that post-training adds new, functionally specialized attention heads that support structured reasoning.

- However, SFT, distillation, and GRPO exhibit rather different behaviors: SFT and distillation add many stable heads (distillation: early–mid layers; SFT: mid–late), while GRPO shows a behavior where few heads are iteratively activated/pruned in sync with reward fluctuations.

- There is also an analysis of the different “thinking modes” in the analysed LLMs. The paper claims that there are no attention heads directly corresponding to “think on/think off”, but that different sets of heads are active under the two settings.

- The paper claims that there is a high-level trade-off in that methods that deepen planning boost complex problem solving but erode basic execution reliability.

**Strengths:**

- The contribution is timely, and while there is previous mechanistic interpretability work looking at reasoning models, this mechanistic analysis of how post-training affects reasoning mechanisms could be an important contribution.

- The emphasis on training dynamics sets this paper apart from previous MI work for reasoning models.

- Also, the analysis of the quite different effects on the model’s circuits of SFT, distillation, and GRPO respectively is interesting and potentially a valuable contribution.

- Similarly, the mechanistic analysis of the different thinking modes seems to be novel.

**Weaknesses:**

- The choice of method is not motivated and there is no discussion of the pros and cons of the selected method.

- The paper does not discuss the issue of polysemanticity at all and whether that is potentially a threat to the validity of the conclusions.

- The circuit discovery method is a bit old-school and it would be relevant to consider more recent methods that are specifically designed to circumvent the problem of polysemanticity, such as circuit discovery methods involving SAEs or Transcoders. Here are a few papers that would be relevant to discuss in this context.

Marks et al. “Sparse Feature Circuits”, (ICLR, 2025).

Caples et al. “Scaling Sparse Feature Circuit Finding to Gemma 9B”, (LessWrong, 2025).

Dunefsky et al. “Transcoders Find Interpretable LLM Feature Circuits”, (NeurIPS, 2024)

- The related work section does not discuss any related work in mechanistic interpretability, and in particular it would be relevant to discuss MI for reasoning specifically. The paper by Cabannes is already cited here, but should be discussed in this context. Other relevant papers include

Dutta et al. “How to think step-by-step: A mechanistic understanding of chain-of-thought reasoning”, (TMLR 2024).

- The last section does not discuss any limitations, despite its title.

- Some of the analysis sections seem more like storytelling rather than rigorously grounded scientific research. To take one instance, the exploration/exploitation discussion on lines 291-294 does not have a solid connection to the empirical results actually presented here: it seems like an example of the sort of “just so” story that MI work is so often criticized for.

- The writing is rather poor throughout the paper, which a minor nuisance which can be fixed in a revised version, but makes the discussion rather hard to follow at some places where the writing cannot be fully deciphered. The abstract has a rather LLM-generated smell to it, so maybe the same LLM could be applied to the rest of the paper to correct weird formulations.

- Please write section 2 more precisely. $N$ seems to be overloaded notation here: it seems to denote both the set of nodes as well as a single node. In the activation patching, spell out more clearly what the output signal refers to.

**Questions:**

- Do you see any threats to the validity of your conclusions? MI research is often on the edge of “reading the tea leaves” – do you see these results as rock solid?

- How do you justify the choice of method?

- In particular, how do you justify not using a circuit discovery method designed to avoid problems related to polysemanticity?

---

> ### Author Response · Authors · 2025-11-19
> **Response to Reviewer bK9w (Part 1/3)**
>
> Thank you for your detailed feedback. Your insights help refine our paper and clarify key aspects.
>
> ---
>
> >- The choice of method is not motivated and there is no discussion of the pros and cons of the selected method.
> >- How do you justify the choice of method?
>
> Our primary goal in this paper is to understand **how different post-training methods reshape attention-head usage in math reasoning**, especially under explicit think on/off control. We therefore needed a circuit method that:
>
> 1. Works directly on large, off-the-shelf models (Qwen, Llama-3.2) without retraining auxiliary models.
> 2. Can be run **across many checkpoints and benchmarks** (AIME, AMC, MATH, GPQA), not just on a few hand-picked prompts.
> 3. Produces **head-level circuits** that can be immediately connected to causal interventions (head ablation and scaling).
>
> For these reasons we chose **EAP-IG** (edge attribution patching with integrated gradients) over heavier feature-level methods: it lets us treat the transformer as a directed graph over standard nodes (input, attention heads, MLPs, output) and assign an importance score to each edge using only forward/backward passes. We use circuits *not* to fully reverse-engineer fine-grained mechanisms, but as a **practical selection mechanism** for “which heads matter” under different post-training regimes, which we then test via ablation and scaling on real benchmarks.
>
> We also spell out the trade-off: this head-level, EAP-IG–based approach is **computationally light and architecture-agnostic**, but coarser than sparse-feature methods. It is well-matched to the scope of our paper (multi-checkpoint, multi-benchmark analysis of post-training), while more detailed topological or feature-level analyses are deferred to future work rather than claimed here.
>
> >- The paper does not discuss the issue of polysemanticity at all and whether that is potentially a threat to the validity of the conclusions.
> >- The circuit discovery method is a bit old-school and it would be relevant to consider more recent methods that are specifically designed to circumvent the problem of polysemanticity, such as circuit discovery methods involving SAEs or Transcoders.
> >- In particular, how do you justify not using a circuit discovery method designed to avoid problems related to polysemanticity?
>
> We agree that **polysemanticity** is a serious concern, and that modern sparse-feature methods (SAEs, Transcoders, Sparse Feature Circuits) are an important advance. In the revision we now discuss these works explicitly and position our approach relative to them.
>
> Our choice *not* to use SAE/transcoder-based circuits is driven by two practical and conceptual constraints:
>
> 1. **Compute and coverage.**
>
>     Methods like Sparse Feature Circuits or Transcoders typically require **training a sparse autoencoder or transcoder for each layer, each component such as MLP, Attn, Res and often each model,** on a representative corpus. Instead, in our setting, we study:
>
>     - Multiple post-training regimes (SFT, distillation, GRPO, think on/off),
>     - Multiple checkpoints per regime, and
>     - Two model families (Qwen and Llama-3.2).
>
>     Training and validating per-layer SAEs for every checkpoint/model combination would have required high computational resources far beyond the scope of this project, and would have forced us to drastically reduce either the number of checkpoints or the number of benchmarks. We judged that, for our specific questions, it was more valuable to maintain **broad coverage across training regimes and tasks** using a lighter-weight circuit method.
>
> 2. **Level of claims.**
>
>     We are careful **not** to claim feature-level semantic purity. Our conclusions are intentionally phrased at a coarse level:
>
>     - certain *heads* become newly important after specific post-training,
>     - ablating or scaling those heads has large, consistent effects on **task-level** math performance, and
>     - these patterns differ between SFT, distillation, GRPO, and think on/off modes.
>
>     We do **not** claim that “this head encodes a single interpretable feature,” only that “this head (possibly polysemantic) is empirically necessary or beneficial for the behaviors we measure.”
>
>     In Section 7 we now call out residual polysemanticity explicitly as a limitation of our methodology.

---

> ### Author Response · Authors · 2025-11-19
> **Response to Reviewer bK9w (Part 2/3)**
>
> 3. **Continued relevance of classical circuits and recency of EAP-IG**
>
>     As we already note in the Related Work, the **classical circuits framework over standard components (attention heads, MLPs, etc.) remains very much active**, and a large body of recent work still adopts node-/edge-level circuits as the basic object of analysis. For example, Hanna et al. explicitly state that “many recent language model interpretability studies have adopted the circuits framework” when motivating their EAP-IG method for finding faithful subgraphs in transformers.
>
>     Likewise, sparse-feature methods such as Sparse Feature Circuits are framed as *extensions* that build on top of these earlier head/neuronal circuits, not as a replacement that renders them obsolete.
>
>     In this context, our choice of **EAP-IG** is not “old-school”: it is a **recent circuit-finding method** introduced at COLM 2024 as an improvement over vanilla edge attribution patching, with better faithfulness to ground-truth circuits.
>
>     EAP-IG is also used as a competitive baseline in subsequent benchmarks like INTERPBENCH and in later mechanistic studies of model reasoning, indicating that it is regarded as a **state-of-the-art edge-based circuit tool**, not a legacy technique [1-3].
>
>     Our work fits into this ongoing “classical circuits + EAP-IG” line: we adopt a standard node-level circuit representation and a modern, faithfulness-oriented circuit finder, while explicitly acknowledging that newer sparse-feature approaches offer a finer granularity that we leave for future work.
>
> 4. **Targeted sparse-feature-circuit analysis**
>
>     To more directly engage with the reviewer’s concern, we additionally perform a **Sparse Feature Circuit analysis** in the revision (see A.5). Here we exploit the fact that full residual-stream SAEs are publicly available through Neuronpedia for **Llama-3.1-8B** and **DeepSeek-R1-Distill-Llama-8B**, and restrict approach to this pair:
>
>     - We **decompose the residual stream** into sparse features using these pre-trained SAEs, while keeping the Attention and MLP blocks **dense** via identity dictionaries.
>     - Prompt is same with EAP-IG, which is based on AIME question and sampled answer.
>     - With this approach, we compute attribution scores, and aggregate at the level of layer–component in Res, Attn, MLP.
>     - For each component we then measure the shift between base and post-trained models using a **symmetric relative difference** in globally normalized importance, and visualize it as a heatmap.
>
>     This analysis yields a clear and interpretable picture that **aligns with our head-level EAP-IG circuits**:
>
>     - We observe a strong **emergence of importance in Layer-0 Attention** in the DeepSeek model, mirroring our finding that early heads become critical after distillation.
>     - **Residual features** show a **progressive strengthening in mid-to-late layers**, suggesting that reasoning increasingly relies on deep, disentangled residual representations.
>     - **MLP blocks** also shift importance toward later layers, though less strongly than the residual stream.
>
>     Because the residual SAEs provide a more monosemantic basis, this serves as an additional check that our conclusions are **not merely artifacts of polysemantic superposition** in the residual stream: the high-level pattern of “early attention + deep residual/MLP reliance” persists under this alternative representation.
>
>     At the same time, this experiment highlights why we do not adopt sparse-feature circuits as our *primary* tool: Attention/MLP remain dense (so we lose head-level resolution), and extending SAEs to **all layers, components, models, and checkpoints** would be computationally prohibitive. We therefore present this SAE-based analysis as a **targeted robustness check** and leave a full, end-to-end sparse-feature-circuit treatment (including SAEs/Transcoders for attention and MLPs) as promising future work.
>
> Overall, EAP-IG thus represents a deliberate **middle ground**: it is more faithful than naive attribution or activation-frequency analysis, but much cheaper and more transferable than per-layer SAEs/Transcoders. We now make this pros/cons trade-off explicit in the paper: our method is *not* designed to solve polysemanticity; instead, we mitigate it indirectly by (i) aggregating circuits across many inputs and (ii) grounding all importance claims in **causal interventions** (head ablation and scaling). A full sparse-feature-circuit treatment would undoubtedly be valuable, but we treat it as an explicit future direction rather than something we quietly approximate.
>
> ---
>
> [1] INTERPBENCH: Semi-Synthetic Transformers for Evaluating Mechanistic Interpretability Techniques, Gupta et al., 2024
>
> [2] Knowledge Circuits in Pretrained Transformers, Yao et al., 2024
>
> [3] Does Time Have Its Place? Temporal Heads: Where Language Models Recall Time-specific Information, Park et al., 2025

---

> ### Author Response · Authors · 2025-11-19
> **Response to Reviewer bK9w (Part 3/3)**
>
> >The related work section does not discuss any related work in mechanistic interpretability, and in particular it would be relevant to discuss MI for reasoning specifically.
>
> We added a dedicated subsection in Related Work that situates our paper within **mechanistic interpretability**, explicitly discussing Cabannes et al., Dutta et al., and related chain-of-thought circuit studies. We clarify that those works focus on reverse-engineering specific reasoning patterns in base models, whereas our contribution is to study **how post-training reshapes head-level circuits and their causal impact on math benchmarks**. We do not claim to match their level of granular mechanism recovery; instead, we connect their insights to our post-training setting.
>
> >The last section does not discuss any limitations, despite its title.
>
> The last section now contains an explicit **Limitations** paragraph, rather than only implicit caveats. We spell out that:
>
> - Our analysis is restricted to math reasoning tasks and a small set of model families (Qwen, Llama-3.2-1B).
> - We work at the **head level**, not at the sparse-feature level, so polysemanticity remains.
> - Our conclusions depend on the choice of benchmarks, prompting scheme (including think on/off templates), and EAP-IG hyperparameters.
>
> We frame our findings as **evidence about these specific settings**, not as universal laws of post-training.
>
> >- the exploration/exploitation discussion on lines 291-294 does not have a solid connection to the empirical results actually presented here: it seems like an example of the sort of “just so” story that MI work is so often criticized for.
> >- do you see these results as rock solid?
>
> We agree that parts of the original exploration/exploitation narrative risked sounding like “reading the tea leaves.” In the revision we will:
>
> - Remove or soften passages where the language went beyond what the plots actually support.
> - Rephrase the exploration/exploitation discussion as a **hypothesis consistent with the observed best-of-k curves**, and we clearly label it as interpretive rather than as an experimentally established mechanism.
> - Ensure that every mechanistic statement is tied to a specific figure or table (e.g., pass@k trends under head ablation/scale-down) rather than free-floating.
>
> We see them as **robust within the scope we study**—they are supported by consistent patterns across multiple checkpoints and benchmarks, and by causal interventions—but still subject to the usual threats in MI: polysemanticity, distribution shift, and model-/task-specific effects.
>
> >The writing is rather poor throughout the paper … makes the discussion rather hard to follow at some places where the writing cannot be fully deciphered.
>
> Even though we noted that some sentences in the main text were later revised with the help of an LLM for style and grammar, we **did not use LLMs to generate the abstract itself**.
>
> The abstract was written by the authors with the aim of compactly conveying a large number of experiments and several interacting insights, and we structured it to foreground our main empirical findings.
>
> That said, we appreciate the concern that parts of the paper—including the abstract—were hard to follow, and we have substantially polished the writing in the revised version: we simplified convoluted sentences, clarified ambiguous formulations, and tightened the connection between claims and figures/tables so that our contributions and empirical evidence are communicated more clearly.
>
> >Please write section 2 more precisely. seems to be overloaded notation here: it seems to denote both the set of nodes as well as a single node. In the activation patching, spell out more clearly what the output signal refers to.
>
> For **Section 2** specifically, we addressed the notation and patching issues you pointed out:
>
> - We clearly distinguish between the set of nodes—for example, $\mathcal{N} = { I, A_{l,j}, M_l, O }$—and individual nodes $n_x, n_y \in \mathcal{N}$. Throughout the paper, each symbol has a single, fixed meaning; no notation is overloaded to denote both a set and an element.
> - In the description of EAP-IG/activation patching, we explicitly spell out what the “output signal” is: the loss (or divergence) on the **target token’s logits** under clean vs. corrupted runs. We clarify that gradients in Eq. (3) are taken with respect to the **pre-activation input of node v** (its residual-stream input), and that the edge score measures how changes at u influence this output signal along the clean–corrupted path.
>
> We hope these clarifications make the technical core of the paper easier to follow.
>
> ---
>
> Thank you for your feedback, which has greatly enhanced our work. We believe these revisions address your comments effectively.

---

> ### Author Response · Authors · 2025-11-24
> **Gentle Reminder by Authors**
>
> Dear Reviewer bK9w,
>
> Thank you for your thoughtful and constructive feedback on our manuscript. We are writing to kindly ask if you have had a chance to review our response to your comments. We have made significant updates to the manuscript based on your valuable suggestions, which we believe have strengthened the paper. We understand the demanding nature of the review process and are grateful for the time and effort you are dedicating to our work.

---

### Official Review · Reviewer_Mczp · 2025-11-01

**Soundness:** 3
**Presentation:** 3
**Contribution:** 3
**Rating:** 6
**Confidence:** 3

**Summary:**

This paper studies the circuit-level changes introduced by various post-training methods - namely SFT, GRPO with two different datasets, and distillation. It investigates attention heads to analyze the sparsity and location of the changes, also accompanied by a training dynamics analysis. Results suggest that different methods introduce most notable changes on different layers, and GRPO introduces the least changes among them, supporting recent work that pictures GRPO as inducing sparse subnetworks training and sparse token distribution reshaping; the analysis of model with a hybrid reasoning (thinking on/off switch) shows that thinking-off induces compensatory mechanisms in the model. Authors also conduct causal analysis by amplifying or suppressing the activation of specific heads, validating their circuit-level findings.

**Strengths:**

S1. Circuit analysis applied to understanding post-training LLM is a novel and evolving area, and the authors tackle important questions with clever and potentially insightful methodology.

S2. The result that methods change heads differently in the layer-wise perspective, reported in Figure 10 and is interesting and important.

S3. Analysis of think on/off regimes provide interesting perspective and might be useful for further research, and is novel to my knowledge.

**Weaknesses:**

W1. Although circuit analysis is reported as the main tool, along with the faithfulness evaluation, the only result reported for circuits is in the Table 1 where emergent heads are listed; in all following results, only the frequency of head activation are used, so circuit analysis might be redundant in the paper. I suggest to cut the circuit-related exposition and focus on increasing the depth of layer-wise change reported in Figure 10, and move this figure to the main body, replacing Table 1.

W2. The activation frequency change metric might be misleading due to resctructurization of the internal computations and emergence of new circuits - e.g., the head L12H1 might play different role in base model and post-trained model, which is confounded by the change of all other parameters and head roles. Therefore, naively comparing frequencies or even importances of specific elements might just be not applicable here. It is important to map elements  (or, if element is decomposed to a circuit of elements or was aggregated from some circuit to a single node, map the circuits) based on their role in the model internal computations, and identify if any element has changed its role, or if some role became redundant/important, or if some roles emerged or disappeared during post-training and have no counterpart.

W3. There are a little conceptual analysis. What is the role of emerged heads and why some heads were deactivated during training? What are the role of the circuits reported in Figures 11 and 12? More concrete case studies would significantly enhance the exposition and the argument, i.e. the results illustrated by clear representative examples, while in current form they do not provide many value and distract from the main conclusions (layer-wise differences, think on/off study, head activation change).

W4. The role of MLP is not studied, although it is the main pattern detection module in transformer; see e.g. [1].

W5. Lines 302-323.5, 368-377.5 are slightly hard to comprehend and would benefit from moving the textual descriptions to plots or tables, while text would contain the main conclusion about what would be presented on a plots.

**Questions:**

Q1. Could you calculate some kind of effect measure between the component importance (computed via attribution patching) on the pre-trained model and post-trained? For example, E(pre, post) = (post - pre) / (pre + post) would be interpreted so that positive (negative) values indicate increased (decreased) importance of that specific module. Another one is simple (post - pre) / pre. If you plot a heatmap for each method, it might be very informative. In figures 3, 6-9 only frequency change is reported, but not the importance; although they might correlate.

Q2. Could you provide a fine-grained case-studies with specific examples of inputs and problem-solving mechanism identified by your analysis? For example, what is the mechanism implemented by circuits reported in figures 11 and 12?

Q3. In lines 355-357 you suggest that circuits would be task-specific, but have you identified such specific circuits that show these differences and task specificity? I can think of the following experiment: given some input question, corrupt specific element (performing patching intervention) and analyse the pass@k behaviour of the model when it tries to solve this problem, also comparing it with baselines (clear model without intervention). Analysis of the failure patterns and responses might reveal the role of each component and allow for task-specific study. This is indeed very expensive, but perhaps might be optimized or re-designed. Do you think that it is feasible to perform such experiment?

[1] Arithmetic Without Algorithms: Language Models Solve Math with a Bag of Heuristics, Nikankin et al., 2025

---

> ### Author Response · Authors · 2025-11-19
> **Response to Reviewer Mczp (Part 1/2)**
>
> Thank you for your detailed and insightful review, which highlights key insights and suggests areas for improvement.
>
> ---
>
> >I suggest to cut the circuit-related exposition and focus on increasing the depth of layer-wise change reported in Figure 10, and move this figure to the main body, replacing Table 1.
>
> We agree that the current presentation under-emphasizes the value of circuit analysis relative to the frequency-based summaries.
>
> In the revision, we re-organize the location of Figure 10 and Table 1. Importantly, circuit analysis is not used only to produce Table 1: the EAP-IG circuits are our *selection mechanism* for identifying candidate heads, which we then probe causally via ablation and scaling in later sections. In other words, the frequency plots are summaries *of circuit-derived head sets*, not independent analyses.
>
> We will make this dependency explicit and streamline the circuit exposition to focus on the pieces that are directly used downstream, while leaving a more detailed layer-wise analysis for future, dedicated work.
>
> >It is important to map elements (or, if element is decomposed to a circuit of elements or was aggregated from some circuit to a single node, map the circuits) based on their role in the model internal computations, and identify if any element has changed its role, or if some role became redundant/important, or if some roles emerged or disappeared during post-training and have no counterpart.
>
> We fully agree that raw activation-frequency changes can be misleading when the internal computation is restructured and heads change roles after training. This is precisely why we do not interpret frequency deltas in isolation: for heads discovered via EAP-IG circuits, we always pair frequency analysis with *causal* interventions (zero ablation and scaling up/down) on task-level benchmarks. This combination allows us to say not only that a head becomes more or less active, but also that its removal or amplification has a measurable effect on math reasoning performance.
>
> At the same time, we see clear value in the reviewer’s suggestion to explicitly map heads—or small circuits—to functional roles (e.g., via structured patching and role alignment across checkpoints or models). A systematic “role-mapping” study of this kind is beyond the scope of the current paper, but we now discuss it as a concrete direction for future work and explicitly flag the interpretational limits of frequency-based metrics in the main text.
>
> >There are a little conceptual analysis. … More concrete case studies would significantly enhance the exposition and the argument, i.e. the results illustrated by clear representative examples, while in current form they do not provide many value and distract from the main conclusions (layer-wise differences, think on/off study, head activation change).
>
> We agree that understanding *what* emergent heads do is as important as showing that they matter. In practice, however, we found that many individual heads exhibit polysemantic behavior and participate in multiple overlapping computations, consistent with prior observations in group-query attention and superposition. This makes it difficult to assign a clean, single “role” to each head without a much more granular feature-level analysis.
>
> Instead, our present work focuses on *groups* of reasoning heads identified by circuits and studies their aggregate impact via ablations and pass@k curves.
>
> >The role of MLP is not studied, although it is the main pattern detection module in transformer.
>
> We appreciate the reminder that MLPs are crucial pattern detectors in transformers. In our circuit extraction, MLP nodes do appear across all configurations, but empirically they were *highly shared* between base and post-trained models and across benchmarks, making it difficult to isolate training-regime–specific effects at the granularity of this paper.
>
> For this reason, we restricted our main analysis to attention heads, where we observed much clearer emergence and turnover patterns. We will defer a MLP-focused analysis to future work that can leverage techniques specifically designed for feature-level interpretability in feedforward layers.
>
> >Lines 302-323.5, 368-377.5 are slightly hard to comprehend and would benefit from moving the textual descriptions to plots or tables, while text would contain the main conclusion about what would be presented on a plots.
>
> We agree that the original prose in these sections was overly dense. In the revision, we have moved the detailed numerical comparisons into explicit tables and simplified the surrounding text to highlight only the main qualitative takeaways (e.g., which head categories lead to the largest performance drops, or how trends differ between SFT and GRPO).

---

> ### Author Response · Authors · 2025-11-19
> **Response to Reviewer Mczp (Part 2/2)**
>
> >Could you calculate some kind of effect measure between the component importance on the pre-trained model and post-trained?
>
> We thank the reviewer for this concrete suggestion. In the revision, we implemented an effect-based comparison between the **attribution-defined importance** of components in the pre-trained and post-trained models. Concretely, we reuse the EAP-IG edge scores defined in Sec. 2 (Eq. 3): for each attention head $(l,h)$, we aggregate the absolute EAP-IG scores of all edges whose source is that head across our AIME and AMC prompts, obtaining a scalar importance $I_{\text{pre}}(l,h)$ for the base model and $I_{\text{post}}(l,h)$ for each post-trained model (DeepSeek-distilled, SFT, and GRPO). After a global normalization per model, we define the reviewer’s symmetric effect measure
>
> $E(l,h);=;\frac{I_{\text{post}}(l,h)-I_{\text{pre}}(l,h)}{I_{\text{post}}(l,h)+I_{\text{pre}}(l,h)+\varepsilon},$
>
> so that positive (negative) values indicate increased (decreased) head importance while keeping the scale bounded. For SFT and GRPO, we compute $I_{\text{post}}$ at each checkpoint and average the resulting $E(l,h)$ over checkpoints to obtain a single “aggregated effect’’ map per training regime; the corresponding heatmaps are included in the appendix.
>
> Empirically, these effect-based heatmaps show strong qualitative agreement with the frequency-based circuit analysis in Figures 3-4 and 7–10. Heads we previously identified as “emergent’’ based on their high appearance frequency in EAP-IG circuits are exactly those that concentrate in regions of large positive $E(l,h)$, while heads that gradually disappear from the circuits across checkpoints tend to have negative effect values. We do not observe qualitatively new clusters of important heads that contradict the frequency view, suggesting that activation frequency is a useful coarse proxy for importance in our *top-circuit* regime, and that our conclusions about which layers and heads are most affected by DeepSeek distillation, SFT, and GRPO remain stable under this stronger, attribution-based notion of component importance.
>
> We clarified this connection and description in A.4, and referred to the effect-measure visualizations when discussing emergent heads.
>
> >Could you provide a fine-grained case-studies with specific examples of inputs and problem-solving mechanism identified by your analysis? For example, what is the mechanism implemented by circuits reported in figures 11 and 12?
>
> Figures 11 and 12 are intended as *prompt-level circuit snapshots*: for a fixed AIME-style problem and prompting condition (think-on vs. think-off), they show which heads receive high EAP-IG scores and how these heads connect across layers. In the current version, we deliberately stop short of claiming a fully worked-out “mechanism” for each circuit, because our analysis operates at the level of whole heads and pass@k performance rather than sparse internal features.
>
> What these figures do provide is a concrete illustration of how the model’s internal attention pathways reorganize once a reasoning trace is enabled: for the same input, think-on prompts route information through a more compact, deeper set of heads, while think-off prompts rely on broader, shallower patterns.
>
> >In lines 355-357 you suggest that circuits would be task-specific, but have you identified such specific circuits that show these differences and task specificity?
>
> We agree that circuits are likely to be task- and input-specific, and we appreciate the concrete experiment the reviewer proposes. Our current analysis already moves partway in this direction: for each benchmark, we identify reasoning-related circuits and then study *pass@k curves under targeted head interventions* (ablation and scaling) while varying the number of sampled solutions. This allows us to observe, for instance, that ablated/scale-down models show a persistently lower best-of-k performance than the unmodified model, indicating a reduced ability to discover diverse correct solutions even when more samples are drawn. In other words, we already use “component-level interventions with pass@k behavior” as a coarse probe of task-specific roles.
>
> That said, we fully agree that combining this with **input-level patching** (e.g., corrupting specific elements of the prompt or intermediate activations and analyzing failure modes for individual problems) would yield a much richer picture of task-specific mechanisms and role changes after training. Such experiments are computationally demanding in our multi-benchmark, multi-checkpoint setting, but they are conceptually aligned with our framework and technically feasible with further optimization.
>
> We discuss this as an explicit avenue for future work: extending our pass@k head-intervention analysis with targeted patching and per-problem error analysis to more directly connect circuits to interpretable failure patterns.
>
> ---
>
> If you have any further question, please feel free to share.

---

> ### Author Response · Authors · 2025-11-24
> **Gentle Reminder by Authors**
>
> Dear Reviewer Mczp,
>
> Thank you for your detailed feedback on our manuscript. We have thoroughly addressed your comments and submitted a revised version for your consideration. We would greatly appreciate it if you could review our responses and the updated manuscript at your earliest convenience. We understand the demanding nature of the review process and are grateful for the time and effort you are dedicating to our work.
>
> Thank you for your time and contribution to improving our work.

---

### Official Review · Reviewer_MbaS · 2025-11-01

**Soundness:** 2
**Presentation:** 1
**Contribution:** 2
**Rating:** 2
**Confidence:** 3

**Summary:**

The paper applies edge attribution patching with integrated gradients (EAP-IG) to trace “reasoning circuits” that emerge after different post-training regimes (SFT, distillation, GRPO). The authors argue that: (1) SFT/distillation introduce a substantial number of attention heads that are absent in the base circuit; (2) GRPO mostly adds a smaller, more targeted set of heads; and (3) for Qwen3-8B, disabling the thinking mode makes the model activate a broader, more redundant circuit, whereas think-on uses a more compact one. The evidence is based on circuit visualizations and head-ablation results on several benchmarks.

**Strengths:**

- Analysis of emergent heads across different post-training regimes raises useful questions for future interpretability work.
- The think-on vs. think-off comparison is interesting and seems worth pursuing further.

**Weaknesses:**

Overall narrative is hard to follow because several experimental details are missing and the structure separates related results.
- How exactly is the EAP-IG score computed? What is taken as the “input of v”? Is the score computed token-wise and then averaged, or do you use a single aggregated embedding?
- Please describe the procedure for constructing “corrupted prompts.”
- There is no figure analogous to Figures 1–2 for the distillation heads; in addition, the current placement of these figures is inconvenient — it would be clearer to group related visualizations together.
- The quantitative analysis in Section 4.2 is difficult to read as text; it would be clearer to present it as a table.
- L359: you state that “when the think process is disabled by predefined <think>\n</think> tokens, the model activates a much larger and more complex set of attention heads,” but in Figure 13 the circuits look comparably sized. Could you provide the explicit list of heads that appear only in the think-off mode?
- You should ablate the same heads in the baseline model and compare the performance changes.

**Questions:**

Please, see weaknesses section.

---

> ### Author Response · Authors · 2025-11-19
> **Response to Reviewer MbaS (Part 1/2)**
>
> We greatly appreciate your valuable feedback, which has been instrumental in improving our paper.
>
> ---
>
> >Overall narrative is hard to follow because several experimental details are missing and the structure separates related results.
>
> In response to your concern that the narrative was hard to follow, we substantially reorganized the paper. We now introduce the circuit formalism and EAP-IG scoring cleanly in the Preliminary (Sec. 2 and Appendix A.3), then circuit construction and ablation methodology (Sec. 3 and Appendix A.1, A.2) before moving to the three main case studies—distillation/SFT (Sec. 4), GRPO (Sec. 5), and think on/off (Sec. 6). Related results and their visualizations are now grouped by training paradigm (e.g., SFT-related figures and tables are colocated with Sec. 4, GRPO with Sec. 5), and we added several quantitative descriptions and tables in the appendix (Tables 4–5). We also added extended experiments (Llama-3.2 re-implementation, Sparse Feature Circuit analysis, additional head-intervention studies, and coverage analysis for think on/off) to make the overall story more coherent and self-contained.
>
> >How exactly is the EAP-IG score computed? What is taken as the “input of v”? Is the score computed token-wise and then averaged, or do you use a single aggregated embedding?
>
> We have clarified the EAP-IG definition in Sec. 2 by giving both the formal equation and a short step-by-step explanation.
>
> Concretely, for each edge $(n_u \to n_v)$ we form a straight-line path between the corrupted and clean activations $z_u'$ and $z_u$ at node $n_u$. We evaluate the model at $m$ interpolation points $(1 - k/m)\, z_u' + (k/m)\, z_u$ for $k = 1, \ldots, m$, and average the gradient of a task-agnostic loss $L$ (for example, a KL divergence between the output logits under the clean activations and under the interpolated activations) with respect to the pre-activation input of node $n_v$. This pre-activation input is the residual-stream vector that $n_v$ receives before applying its operation, that is, the sum of all parent outputs at that node. We then take a dot product between this integrated gradient at node $n_v$ and the input difference at $n_u$, $\Delta z_u = z_u - z_u'$, to obtain a scalar edge score $s(n_u \to n_v)$ for that specific example (see Eq. 3). This path is defined over the full token sequence in the residual stream; we do not pool tokens into a single embedding. Instead, each edge's score is computed at its own (layer, head, token) coordinates and then aggregated across many examples (for example, by averaging absolute scores) when ranking edges and selecting circuits.
>
> Additional explanation is added in the revised version of Appendix A.3.
>
> >Please describe the procedure for constructing “corrupted prompts.”
>
> We now describe the clean/corrupted prompt construction explicitly in Sec. 3.1 and Appendix A.2.
>
> For each benchmark instance (e.g., an AIME problem), we fix the **same problem statement** and attach different model continuations to build a clean–corrupted pair. When we construct circuits for the baseline model, the *clean* prompt uses the baseline’s own explanation-style answer (e.g., “To determine …”, “We’ll use Python to …”), while the reasoning model’s `<think>`-style continuation serves as the *corrupted* version. For reasoning models, the roles are reversed: the clean prompt uses a sampled reasoning trace immediately after the `<think>` token (e.g., “Okay, so I have this problem …”), and the baseline’s explanation becomes the corrupted prompt.
>
> This symmetric design yields tightly controlled pairs that isolate the effect of “reasoning vs. non-reasoning” behavior on internal circuits, and we now spell this out with concrete examples in the appendix.
>
> >There is no figure analogous to Figures 1–2 for the distillation heads; in addition, the current placement of these figures is inconvenient — it would be clearer to group related visualizations together.
>
> Because we do not have access to intermediate checkpoints of the DeepSeek-R1 distillation process, we cannot construct a temporal cohort plot analogous to overview figure or Figures 3–4 for that model; we can only analyze the **final distilled checkpoint**.
>
> We therefore focus on the aggregate circuit structure (Fig. 2) and on causal evidence via ablation (Table 1), which show that distillation introduces a large set of new heads, especially in early-to-mid layers, whose removal consistently harms performance. To address the presentation issue, we have reorganized figures so that distillation and SFT analyses (Fig. 3 and Tables 1, 3–5) are grouped in Sec. 4, GRPO analyses (Fig. 4–5 and associated tables) in Sec. 5, and think on/off visualizations (Fig. 6 and Table 2) in Sec. 6, avoiding the previous interleaving of GRPO and SFT figures.

---

> ### Author Response · Authors · 2025-11-19
> **Response to Reviewer MbaS (Part 2/2)**
>
> >The quantitative analysis in Section 4.2 is difficult to read as text; it would be clearer to present it as a table.
>
> We agree and have moved the key quantitative results from prose into explicit tables. In particular, Table 5 now summarizes the SFT and GRPO head-intervention experiments that were previously described only in text: ablation of mid-to-late SFT heads, and scaling up/down GRPO heads in the baseline model. Similarly, Tables 3–4 collect circuit-level emergent head lists and ablation outcomes for Qwen2.5-Math across model sizes and training regimes. We believe this tabular presentation makes the trade-offs (e.g., gains on MATH vs. drops on AIME/AMC) much easier to parse at a glance.
>
> | Model | Method | AIME'24 | AMC | MATH |
> | --- | --- | --- | --- | --- |
> | Qwen2.5-Math-1.5B | No Ablation | 13.3 | 38.5 | 56.0 |
> |  | Ablation with SFT Heads | 0.00 | 0.05 | 0.10 |
> |  | Scale Up with SFT Heads | 0.00 | 37.3 | 58.2 |
> |  | Scale Down with GRPO GSM8K Heads | 3.33 | 42.1 | 63.0 |
> |  | Scale Up with GRPO GSM8K Heads | 3.33 | 30.1 | 60.2 |
>
> >L359: you state that “when the think process is disabled by predefined [object Object]\n[object Object] tokens, the model activates a much larger and more complex set of attention heads,” but in Figure 13 the circuits look comparably sized. Could you provide the explicit list of heads that appear only in the think-off mode?
>
> We apologize for the confusion. Figure 13 shows a **single AIME circuit** for illustration, where the size difference is less pronounced, whereas our claim was based on aggregating circuits across all benchmarks (AIME’24/25, AMC, GPQA, MATH).
>
> We provide an explicit example list of attention heads that appear exclusively in the think-off circuits. Across 326 distinct heads observed, 110 are unique to think-off circuits and never appear in think-on circuits. For instance, the list includes heads such as L0.H27, L1.H19, L10.H19, L10.H23, L10.H31, L10.H5, L11.H0, L11.H2, L11.H31, L11.H4, and others.
>
> We connect this list directly to the quantitative interventions in Table 2, where ablating or scaling down these think-off-only heads changes performance in the way described in the main text.
>
> >You should ablate the same heads in the baseline model and compare the performance changes.
>
> We have added exactly this comparison. In addition to ablating emergent “reasoning heads” in the post-trained models, Table 4 now reports ablations of the **corresponding base-model heads** on the same benchmarks for Qwen2.5-Math-1.5B and 7B.
>
> Interestingly, the trend reverses between DeepSeekR1-Distill and Qwen2.5-Math: in the distilled model, ablating emergent reasoning heads is most damaging, whereas in Qwen2.5-Math, ablating certain base-model heads can hurt more or even *improve* performance, indicating that some pre-existing heads become confusing once new reasoning machinery is installed.
>
> This baseline-vs-reasoning ablation comparison directly addresses your suggestion and strengthens our claim that the identified heads are functional, training-regime–specific units rather than a universal pool.
>
> | Model | Method | AIME'24 | AIME'25 | GPQA | AMC |
> | --- | --- | --- | --- | --- | --- |
> | Qwen2.5 Math-1.5B | No Ablation | 13.3 | 4.73 | 9.74 | 38.5 |
> |  | Ablation with Reasoning Heads | 9.01 | 4.58 | 7.82 | 35.6 |
> |  | Ablation with Base Model Heads | 8.33 | 4.63 | 9.79 | 34.2 |
> |  | Ablation with TriviaQA Heads | 0.05 | 0.00 | 5.38 | 3.42 |
> | Qwen2.5 Math-7B | No Ablation | 13.3 | 10.0 | 15.1 | 32.5 |
> |  | Ablation with Reasoning Heads | 6.67 | 10.0 | 20.2 | 43.3 |
> |  | Ablation with Base Model Heads | 23.3 | 3.33 | 15.6 | 43.3 |
> |  | Ablation with TriviaQA Heads | 20.0 | 10.0 | 16.1 | 37.3 |
>
> ---
>
> We hope these responses address your questions and concerns. Your feedback has been essential in improving our work, and we look forward to refining it further. Thank you for your review.

---

> ### Author Response · Authors · 2025-11-24
> **Gentle Reminder by Authors**
>
> Dear Reviewer MbaS,
>
> Thank you for your thoughtful and constructive feedback on our manuscript. We have thoroughly addressed your comments and submitted a revised version for your consideration. We would greatly appreciate it if you could review our responses and the updated manuscript at your earliest convenience. As the discussion period is progressing, we would be happy to answer any further questions or provide additional clarifications if needed. Thank you for your time and contribution to improving our work.

---

> > ### Comment · Reviewer_MbaS · 2025-11-26
> >
> > The authors have answered my original questions, added several new experiments, and substantially clarified the methodology and structure. This resolves my main concerns, so I will update Presentation from 1 to 3, Soundness from 2 to 3, and Overall score from 2 to 6.
> >
> > However, I remain unconvinced by the strength of the causal interpretation of the identified reasoning heads. Ablating the same head sets produces markedly different effects across benchmarks and between the Qwen and LLaMA experiments. This suggests substantial dependence on the specific experimental setup and makes it difficult to draw general, mechanism-level conclusions about these attention heads.

---

> > > ### Author Response · Authors · 2025-11-27
> > > **Official Comment by Authors**
> > >
> > > Thank you very much for your thoughtful feedback and for taking the time to reassess our work after the revisions.
> > >
> > > We fully understand and appreciate your remaining concern about the strength of the causal interpretation of the identified reasoning heads. We agree that ablating the same head sets can have markedly different effects across benchmarks and between different models indicates a  dependence on the specific experimental setup.
> > >
> > > Here, we see this setup dependence as an additional insight. Even within a single series of reasoning head set, different heads likely implement **distinct functional patterns**: some may be particularly aligned with step-by-step, MATH-style problem solving, while others may generalize poorly to AMC-style problems and inadvertently bias the model toward less suitable reasoning strategies. The variability in ablation outcomes across tasks and families therefore present a richer structure—that **reasoning heads are not homogeneous**, and that the same head can be helpful in one reasoning regime while becoming detrimental in another.
> > >
> > > Moving forward, we plan to investigate this finer-grained functionality with an eye toward improving post-training itself. Rather than treating circuit analysis as an end in itself, we see it as a practical signal for designing **head-aware post-training policies**: for example, using per-head effect estimates to identify heads that are consistently helpful or harmful for specific task families, and then exploring training schemes that selectively reinforce or regularize such heads during post-training. Our future goal is to test whether this kind of head-sensitive objective can reduce the current trade-offs between benchmarks—by dynamically strengthening heads that are essential for more difficult reasoning tasks—while remaining grounded in conservative, empirically validated head-level effects rather than in overly strong mechanistic claims.
> > >
> > > Thank you again for raising this point—your critique directly sharpened both our interpretation and our research agenda.

---

### Author Response · Authors · 2025-11-28
**General Response to All Reviewers**

Dear Reviewers and Chairs,

We thank all reviewers for their feedback and for emphasizing that our work offers a **novel and timely mechanistic perspective** on how post-training and think-on/off behavior reshape model circuits, supported by training-dynamics analyses with causal ablation evidence (@MbaS, @Mczp, @bK9w, @yRKK, @hGiC). Reviewers also noted that the revised version is more **careful, better contextualized** within prior work, and **empirically stronger**, with additional analyses that deepen the structural picture and clarify limitations, which we greatly appreciate (@MbaS, @hGiC).

Across reviews, four themes recurred: (i) clarity and missing details, (ii) motivation and limitations, (iii) ablation design and interpretation, and (iv) generalization. Below we explain how the revision addresses each.

---

### 1. Clarity, Structure, and Missing Details

While retaining the same high-level order, we **substantially revised each stage**: preliminaries now clearly introduce circuits and EAP-IG, methods give a step-by-step circuit/ablation pipeline, and figures/tables are regrouped by training regime with explicit captions and previously text-only numbers moved into tables. (@MbaS, @Mczp, @bK9w, @yRKK)

We now **precisely define EAP-IG** (path, loss signal, gradient target, “input of v”), describe clean–corrupted prompt construction for all benchmarks, and clarify how think-on/off templates determine where we compute circuits, so that the pipeline can be reconstructed without guessing missing details. (@MbaS, @bK9w, @yRKK)

We also add a **mechanistic-interpretability subsection** to explicitly discuss our focus on attention heads rather than MLP, and feature-level. We frame this scope—along with fine-grained case studies for tracing behavior and designing head-aware training policies—as intentional limitations and future work, rather than omitted details. (@bK9w, @Mczp, @yRKK, @hGiC)

### 2. Method Choice, Polysemanticity, and Scope

We chose EAP-IG circuits because they **scale to multiple post-training regimes and checkpoints** on large models using only forward/backward passes, whereas per-layer SAE/Transcoder training would be prohibitively expensive in our multi-regime, multi-checkpoint setting. In this sense, we use circuits as a practical selection mechanism for “which heads to intervene on.” (@bK9w, @Mczp)

We **explicitly acknowledge head-level polysemanticity** and only claim coarse-grained effects—which heads gain or lose importance under different regimes and how their removal/amplification affects performance—rather than monosemantic roles. (@bK9w, @Mczp)

As robustness checks, we add (i) **effect measure comparing head importance** between base and post-trained models, (ii) **residual-SAE–based feature-level analysis** on Llama-3.1/DeepSeek-R1 and (iii) **Llama-3.2-1B SFT/GRPO experiments**, all of which show early-attention and deep-layer shifts consistent with our Qwen results. (@Mczp, @bK9w, @yRKK, @hGiC)

### 3. Experimental Design, Ablations, and Think On/Off

Sec. 2 and App. A.3 now specify the interpolation path, loss, gradient target, and aggregation over tokens/examples, and Sec. 3 and App. A.2 describe how we extract benchmark-specific circuits for each benchmark, addressing questions about the “output signal” and the role of each node in the computation graph. (@MbaS, @Mczp, @bK9w)

We spell out clean–corrupted pair construction (reasoning vs. non-reasoning continuations, with roles reversed for baseline vs. reasoning models) and explain how the Qwen3 chat template toggles `<think>` on/off and which tokens are used for circuits in each case. (@MbaS, @yRKK)

We extend ablations to reasoning-only, base-only, and TriviaQA-only heads and their unions, showing **heterogeneous head populations**: some heads help, some hurt at small pass@k, but **jointly ablating all circuit heads strongly degrades performance**, addressing concerns about “confuser” heads and surprising gains. We also clarify that “TriviaQA heads” appear only in TriviaQA circuits, not math circuits. (@MbaS, @yRKK, @hGiC)

We justify zeroing ablation as the intervention, contrast it with mean ablation, and in the think on/off study pair these interventions with pass@k curves, where weakening reasoning heads consistently lowers best-of-k performance. We now present the exploration/exploitation story—and the associated “over-thinking” trade-offs where mechanisms that help complex problems can hurt simpler ones—as a hypothesis consistent with these curves, not as a proven mechanism. (@yRKK, @bK9w, @hGiC)

---

We again thank all reviewers for their insightful comments. We also deeply appreciate the Committee's dedication amidst recent disruptions. While we regret that the suspension curtailed our positive ongoing discussions before final confirmation, we respectfully ask the AC to consider our comprehensive rebuttal which we believe the revised version is clearer, more robust, and better aligned with the reviews.

---

### Meta-Review · Area_Chair_t4ru · 2026-01-06

**Summary:**

This paper investigates the mechanistic dynamics of Large Reasoning Models (LRMs) during post-training. Using Edge Attribution Patching with Integrated Gradients (EAP-IG), the authors analyze how SFT, Distillation, and GRPO modify internal circuits (specifically attention heads) in Qwen and Llama models. The authors report that SFT and distillation tend to cumulatively add stable reasoning heads, whereas GRPO operates via a dynamic search process. They also analyze the trade-offs of explicit <think> tokens, suggesting that "thinking" enables more efficient circuits compared to the compensatory mechanisms required when thinking is disabled.

**Reviewer Concerns:**

Addressed Concerns:
- Clarity: The authors successfully reorganized the manuscript to clarify the experimental setup (EAP-IG computation, prompt construction), satisfying Reviewers MbaS and Mczp.
- Generalization: The inclusion of Llama-3.2 experiments addressed the concern that findings were specific only to Qwen.

Outstanding Concerns:
- Methodological Novelty and Depth: The paper applies an existing method (EAP-IG) to a new setting without proposing novel technical contributions. As noted by Reviewer bK9w, the reliance on head-level circuits without fully addressing polysemanticity (despite a supplementary SAE analysis) limits the reliability of the mechanistic claims.
- Causal Strength: Reviewer MbaS remained unconvinced by the strength of the causal interpretation, noting that ablating the same head sets produced markedly different effects across benchmarks, suggesting the "reasoning heads" may not be as functionally distinct as claimed.
- Ablation Baselines: Reviewer YRKK raised valid concerns regarding the use of zero-ablation versus mean-ablation, which casts some doubt on whether the observed performance drops are due to mechanism removal or distribution shifts.
- Lack of Topological Analysis: While acknowledged by the authors, the absence of a deeper topological analysis (centrality, motifs) limits the insights to simple activation frequencies, which Reviewer hGiC noted scratches the surface of circuit analysis.

**Reviewer Scores:**

- Reviewer MbaS (Score: 2): Raised from 2 to 6 explicitly.
- Reviewer Mczp (Score: 6): Maintained 6.
- Reviewer hGiC (Score: 6): Maintained 6.
- Reviewer bK9w (Score: 4): Would likely remain at 4  as the fundamental critique regarding polysemanticity and the preference for SAEs over EAP-IG remains unaddressed by the core methodology.
- Reviewer YRKK (Score: 2): Would likely remain at 2  due to fundamental disagreement with the zero-ablation strategy used to define head importance.

---

### Decision · Program_Chairs · 2026-01-26

Reject